The clinical diagnosis of Achilles tendinopathy: a scoping review

http://orcid.org/0000-0002-2281-0464 Matthews Wesley 1 wesley.matthews@student.bond.edu.au
http://orcid.org/0000-0001-6848-6842 Ellis Richard 2 3
http://orcid.org/0000-0001-7773-0253 Furness James 1
Hing Wayne A. 1
1 Bond Institute of Health and Sport, Faculty of Health Sciences and Medicine, Bond University , Gold Coast, Queensland , Australia
2 Active Living and Rehabilitation: Aotearoa New Zealand, Health and Rehabilitation Research Institute, Faculty of Health and Environmental Sciences, Auckland University of Technology , Auckland , New Zealand
3 Department of Physiotherapy, School of Clinical Sciences, Faculty of Health and Environmental Sciences, Auckland University of Technology , Auckland , New Zealand
Okpala Charles
Electronic publication date: 2021 Sep 28
Publication date: 2021
Volume: 9
Electronic Location ID: e12166
Received 2021 May 6; Accepted 2021 Aug 25
Copyright: © 2021 Matthews et al.
Copyright year: 2021
Copyright holder: Matthews et al.
License: This is an open access article distributed under the terms of the Creative Commons Attribution License, which permits unrestricted use, distribution, reproduction and adaptation in any medium and for any purpose provided that it is properly attributed. For attribution, the original author(s), title, publication source (PeerJ) and either DOI or URL of the article must be cited.
License URL: https://creativecommons.org/licenses/by/4.0/

Keywords: Tendinopathy, Diagnosis, Achilles, Tendon, Tendinosis, Tendinitis

Funding: Australian Government Research Training Program Scholarship This research was supported by an Australian Government Research Training Program Scholarship. The funders had no role in study design, data collection and analysis, decision to publish, or preparation of the manuscript.

==============================
Background

Achilles tendinopathy describes the clinical presentation of pain localised to the Achilles tendon and associated loss of function with tendon loading activities. However, clinicians display differing approaches to the diagnosis of Achilles tendinopathy due to inconsistency in the clinical terminology, an evolving understanding of the pathophysiology, and the lack of consensus on clinical tests which could be considered the gold standard for diagnosing Achilles tendinopathy. The primary aim of this scoping review is to provide a method for clinically diagnosing Achilles tendinopathy that aligns with the nine core health domains.

Methodology

A scoping review was conducted to synthesise available evidence on the clinical diagnosis and clinical outcome measures of Achilles tendinopathy. Extracted data included author, year of publication, participant characteristics, methods for diagnosing Achilles tendinopathy and outcome measures.

Results

A total of 159 articles were included in this scoping review. The most commonly used subjective measure was self-reported location of pain, while additional measures included pain with tendon loading activity, duration of symptoms and tendon stiffness. The most commonly identified objective clinical test for Achilles tendinopathy was tendon palpation (including pain on palpation, localised tendon thickening or localised swelling). Further objective tests used to assess Achilles tendinopathy included tendon pain during loading activities (single-leg heel raises and hopping) and the Royal London Hospital Test and the Painful Arc Sign. The VISA-A questionnaire as the most commonly used outcome measure to monitor Achilles tendinopathy. However, psychological factors (PES, TKS and PCS) and overall quality of life (SF-12, SF-36 and EQ-5D-5L) were less frequently measured.

Conclusions

There is significant variation in the methodology and outcome measures used to diagnose Achilles tendinopathy. A method for diagnosing Achilles tendinopathy is proposed, that includes both results from the scoping review and recent recommendations for reporting results in tendinopathy.

Introduction

Achilles tendinopathy describes the clinical presentation of pain localised to the Achilles tendon and associated loss of function with tendon loading activities (De Vos et al., 2021; Millar et al., 2021). However, clinicians display differing approaches to the diagnosis of Achilles tendinopathy due to inconsistency in the clinical terminology, an evolving understanding of the pathophysiology, and the lack of consensus on clinical tests which could be considered the gold standard for diagnosing Achilles tendinopthy (De Vos et al., 2021; Millar et al., 2021; Docking, Ooi & Connell, 2015; Cook et al., 2016). Conversely, when describing the clinical condition of persistent pain and dysfunction of the Achilles tendon in relation to mechanical loading, consensus agreement has identified the preferred terminology to be ‘tendinopathy’ rather than other common terms such as ‘tendinitis’ and ‘tendinosis’ (Scott et al., 2020). However, the consensus agreement for terminology does not provide a clear criteria with which to diagnose Achilles tendinopathy (De Vos et al., 2021).

Additionally, when considering the diagnosis of Achilles tendinopathy, distinctions can be made between the diagnosis of tendinopathy and clinical diagnosis of Achilles tendinopathy. As described by Aggarwal et al. (2015), a diagnosis is based off a broad set of signs and symptoms to reflect all the potential features and severity of a pathology. Whereas, a clinical diagnosis of Achilles tendinopathy requires a specific set of signs, symptoms and tests to define a homogenous group of patients across studies and geographical regions (Aggarwal et al., 2015). In the case of Achilles tendinopathy, the diagnosis of Achilles tendinopathy is determined by the presentation of pain localised to the Achilles tendon and associated loss of function with tendon loading activities (De Vos et al., 2021; Millar et al., 2021). However, this broad description may include other pathological disease processes such as retrocalcaneal bursitis, complete or partial rupture of the Achilles, tarsal tunnel syndrome, neuroma/neuritis of the sural nerve, rupture posterior tibial tendon, or arthritic conditions of the ankle that need to be differentially diagnosed (Hutchison et al., 2013). Thus, it becomes relevant to understand the process to determine a clinical diagnosis of Achilles tendinopathy.

The clinical diagnosis of Achilles tendinopathy is predominantly derived from patient history, patient reported load related pain, and pain provocation tests (Millar et al., 2021). Patient history, localised Achilles tendon pain and pain on palpation are considered key to diagnosing Achilles tendinopathy (De Vos et al., 2021; Millar et al., 2021) and can all be assessed reliably (Hutchison et al., 2013). Additional pain provoking tests; such as the single leg heel raise, hop test, Royal London Hospital Test or Painful Arc Sign; have been suggested as useful to confirm a clinical diagnosis of Achilles tendinopathy (Millar et al., 2021; Hutchison et al., 2013; Reiman et al., 2014). However, many leading researchers disagree on the which clinical tests are essential to diagnose Achilles tendinopathy (De Vos et al., 2021). Conversely, it is agreed that uniform diagnostic criteria would be useful in identifying possible subclassifications of Achilles tendinopathy and thus improving tailored individual treatment programmes or monitoring patient progress (De Vos et al., 2021).

Recently, Vicenzino et al. (2020) identified nine core health domains in tendinopathy following consensus agreement from both health care practitioners and patients. These included patient rating of overall condition, pain on activity or loading, participation, function, psychological factors, disability, physical function capacity, quality of life, and pain over a specified timeframe (Vicenzino et al., 2020). An overview of the nine core health domains of tendinopathy (Vicenzino et al., 2020) are presented in Table 1. Using the determined core health domains, specific measures will need to be identified specific to Achilles tendinopathy (Vicenzino et al., 2020). The introduction of the nine core health domains in tendinopathy (Vicenzino et al., 2020) in addition to previously identified gaps in the literature, including; a lack of consistency in terminology used to diagnose Achilles tendinopathy (De Vos et al., 2021; Scott et al., 2020), lack of a consensus on the clinical diagnosis of Achilles tendinopathy (De Vos et al., 2021), and the need for a uniform method with which to clinically diagnose Achilles tendinopathy (De Vos et al., 2021). Thus there is a requirement to identify the methods with which these gaps can be addressed and allow for greater consistency in the clinical diagnosis of Achilles tendinopathy in both research and clinical practice.

Table 1 The nine core health domains of tendinopathy as recommended by Vicenzino et al. (2020).

Domain	Description	Example	
Patient rating of overall condition	A single assessment numerical evaluation	0–100%	
Pain on activity or loading	Patient reported intensity of pain during a tendon loading activity.	VAS, NRS	
Participation	Patient rating of participation levels in sport or engagement across other areas.	Tegner Activity Scale	
Function	Patient rating of function and not referring to the intensity of their pain.	Patient Specific Function Scale	
Psychological factors	Patient rating of psychological impact (e.g. Pain self efficacy, kinesiophobia, catastrophisation) .	PCS	
Disability	Scores from a combination of patient rated pain and disability due to pain in relation to tendon specific loading activities	VISA-A	
Physical function capacity	The quantitative measures of physical tasks such as number of hops, number of squats and dynamometry.	Single leg heel raise	
Quality of life	Patient rating of general wellbeing	EQ-5D	
Pain over a specified time	Patient reported intensity of pain over a specified time period (e.g. morning, night, 24 h).	VAS, NRS	
Note:

VAS, visual analogue scale; NRS, numerical rating scale; PCS, pain catastrophisation scale; VISA, Victorian Institute of Sport Assessment; EQ-5D, EuroQol-5 dimension.

Therefore, the primary aim of this scoping review is to provide a method for clinically diagnosing Achilles tendinopathy that aligns with the nine core health domains. In order to achieve this, specific objectives have been determined that include: (1) identifying the most common clinical tests used to diagnose Achilles tendinopathy, (2) identifying the most common outcome measures used to assess Achilles tendinopathy, and (3) summarising the studies to date.

Methodology

Study design

A scoping review was conducted to synthesise available evidence on the clinical diagnosis and clinical outcome measures of Achilles tendinopathy. Due to the wide-ranging nature of the topic, a scoping review was used to facilitate the collection and charting of evidence with the aim of identifying key themes, knowledge gaps and types of evidence currently available. Figure 1 provides an overview of the overall study design and process to answer the primary aim and specific objectives.

Figure 1 Overall study design.

Search strategy

A single researcher (WM) completed a literature search to identify, screen and select studies in accordance with the Preferred Reporting Items for Systematic Reviews and Meta-Analysis Extension for Scoping Reviews (PRISMA-ScR) (Tricco et al., 2018). A detailed, multistep search of PubMed, CINAHL, ProQuest and SPORTDiscus was conducted between May 2020 and July 2020, before being updated in April 2021. In addition to the electronic database search, reference lists from included articles were reviewed for additional articles. To ensure a broad search, key words were truncated to allow for variations in spelling and combined using Boolean operators in addition to the use of MeSH terms to allow for review of all relevant articles. The full electronic search for the PubMed database is provided in Table 2.

Table 2 Electronic database search strategy example.

Database	Search strategy	Results	
PubMed	(“tendineous”[All Fields] OR “tendinopathy”[MeSH Terms] OR “tendinopathy”[All Fields] OR “tendinitis”[All Fields] OR “tendons”[MeSH Terms] OR “tendons”[All Fields] OR “tendinous”[All Fields] OR (“tendinopathy”[MeSH Terms] OR “tendinopathy”[All Fields] OR “tendinosis”[All Fields]) OR (“tendinopathy”[MeSH Terms] OR “tendinopathy”[All Fields] OR “tendinopathies”[All Fields]) OR (“tendinopathy”[MeSH Terms] OR “tendinopathy”[All Fields] OR “tendonopathy”[All Fields]) OR (“tendinopathy”[MeSH Terms] OR “tendinopathy”[All Fields] OR “tendonitis”[All Fields] OR “tendon s”[All Fields] OR “tendonous”[All Fields] OR “tendons”[MeSH Terms] OR “tendons”[All Fields] OR “tendon”[All Fields]) OR (“tendinopathy”[MeSH Terms] OR “tendinopathy”[All Fields] OR “tendonosis”[All Fields])) AND (“diagnosable”[All Fields] OR “diagnosi”[All Fields] OR “diagnosis”[MeSH Terms] OR “diagnosis”[All Fields] OR “diagnose”[All Fields] OR “diagnosed”[All Fields] OR “diagnoses”[All Fields] OR “diagnosing”[All Fields] OR “diagnosis”[MeSH Subheading]) AND (“achiles”[All Fields] OR “achille”[All Fields] OR “achille s”[All Fields] OR “achilles tendon”[MeSH Terms] OR (“achilles”[All Fields] AND “tendon”[All Fields]) OR “achilles tendon”[All Fields] OR “achilles”[All Fields])	7,162	

Eligibility criteria

Methods for data extraction specific to scoping reviews were informed by the Population-Concept-Context framework as recommended by the Joanna Briggs Institute (JBI) Reviewer’s Manual (Peters et al., 2020). Population was defined as any person clinically diagnosed with Achilles tendinopathy regardless of location (insertional or midportion). Concept included any study reporting on the methods used to clinically diagnose Achilles tendinopathy including subjective measures, objective measures and outcome measures. Context included all periods of time, outcomes, comparators, follow-up, rehabilitation settings and duration and type of intervention.

Eligible articles were full-text and included original research, reviews, scoping reviews, systematic reviews, meta-analyses, case-series and clinical commentaries. Studies were included if they provided adequate information on the method of clinical diagnosis (either subjective measures, objective measures or both subjective and objective measures), and clinical outcome measures used. Studies were excluded if they were non-English, had no description of clinical diagnosis, not specific to Achilles tendinopathy or included asymptomatic Achilles tendon states only.

Data extraction and synthesis

WM extracted data from publications meeting the inclusion criteria into an Excel spreadsheet. Data extraction, grouping and plotting were performed by WM in line with previously published recommendations (Peters et al., 2020), where extracted data included author, year of publication, participant characteristics, methods for diagnosing Achilles tendinopathy and outcome measures. Data was extracted in tabular and graphical forms with results grouped by study design and categorised according to the hierarchy of evidence (Daly et al., 2006; Evans, 2003; Merlin, Weston & Tooher, 2009). Diagnostic criteria were presented in tabular form including year of publication, population, subjective and objective measures. Terminology and outcome measures were presented in graphical form with terminology grouped by publication year and outcome measures grouped by purpose of measure (disability, pain, psychological, quality of life).

Following data extraction, data synthesis was performed according to a previously published methodological framework (Thomas & Harden, 2008). Data was synthesised into the following categories: (1) subjective measures, (2) objective measures and (3) outcome measures. Results were plotted according to publication date, terminology, study design and clinical diagnostic measures. Results were then compared to the nine core health domains of tendinopathy (Vicenzino et al., 2020) to identify areas of overlap and gaps in the current evidence. Studies could be allocated to multiple groups. Quality appraisal was not required as per recommended methodology for scoping reviews (Peters et al., 2020; Arksey & O’Malley, 2005).

Results

Selection of sources of evidence

The search results are displayed in the PRISMA Flow Diagram (Fig. 2). The search strategy generated 11,561 results with two further results identified via reference list searching. Following duplicate removal and title and abstract screening, 554 full-text articles were reviewed for inclusion in the study. Of these, 395 were excluded for the following reasons: 240 provided insufficient information on the method of diagnosing Achilles tendinopathy, 11 assessed asymptomatic Achilles tendons only, 46 did not have access to the full text, 52 were not in English and 47 were not specific to Achilles tendinopathy. Thus, 159 articles (Millar et al., 2021; Hutchison et al., 2013; Reiman et al., 2014; Abate & Salini, 2019; Aiyegbusi, Tella & Sanusi, 2020; Aldridge, 2004; Alfredson, 2003; Alfredson & Cook, 2007; Alfredson & Spang, 2018; Aronow, 2005; Asplund & Best, 2013; Azevedo et al., 2009; Bains & Porter, 2006; Barge-Caballero et al., 2008; Barker-Davies et al., 2017; Baskerville et al., 2018; Benazzo, Todesca & Ceciliani, 1997; Benito, 2016; Bhatty, Khan & Zubairy, 2019; Bjordal, Lopes-Martins & Iversen, 2006; Boesen et al., 2017; Borda & Selhorst, 2017; Brown et al., 2006; Carcia et al., 2010; Cassel et al., 2018; Chazan, 1998; Cheng, Zhang & Cai, 2016; Chester et al., 2008; Chimenti et al., 2016; Chimenti et al., 2017; Chimenti et al., 2020; Cook, Khan & Purdam, 2002; Coombes et al., 2018; Courville, Coe & Hecht, 2009; Creaby et al., 2017; Crill, Berlet & Hyer, 2014; De Jonge et al., 2011; De Marchi et al., 2018; Den Hartog, 2009; Divani et al., 2010; Docking et al., 2015; Duthon et al., 2011; Ebbesen et al., 2018; Eckenrode, Kietrys & Stackhouse, 2019; Feilmeier, 2017; Finnamore et al., 2019; Florit et al., 2019; Fredericson, 1996; Furia & Rompe, 2007; Gärdin et al., 2016; Gatz et al., 2020; Habets et al., 2017; Hasani et al., 2020; Hernández-Sánchez et al., 2018; Holmes & Lin, 2006; Horn & McCollum, 2015; Hu & Flemister, 2008; Hutchison et al., 2011; Irwin, 2010; Järvinen et al., 2001; Jayaseelan, Weber & Jonely, 2019; Jewson et al., 2017; Jowett, Richmond & Bedi, 2018; Jukes, Scott & Solan, 2020; Kader et al., 2002; Karjalainen et al., 2000; Khan et al., 2003; Knobloch, 2007; Knobloch et al., 2007; Knobloch et al., 2008; Kragsnaes et al., 2014; Krogh et al., 2016; Lakshmanan & O’Doherty, 2004; Leung & Griffith, 2008; Lohrer & Nauck, 2009; Longo et al., 2009; Longo, Ronga & Maffulli, 2009; Maffulli, Giai Via & Oliva, 2015; Maffulli, Giuseppe Longo & Denaro, 2012; Maffulli & Kader, 2002; Maffulli et al., 2003; Maffulli et al., 2011; Maffulli et al., 2020; Maffulli et al., 2012; Maffulli et al., 2008; Maffulli et al., 2019; Maffulli, Sharma & Luscombe, 2004; Maffulli, Via & Oliva, 2014; Maffulli et al., 2008; Mafi, Lorentzon & Alfredson, 2001; Magnussen, Dunn & Thomson, 2009; Mansur et al., 2019; Mansur et al., 2017; Mantovani et al., 2020; Martin et al., 2018; Mayer et al., 2007; McCormack et al., 2015; McShane, Ostick & McCabe, 2007; Murawski et al., 2014; Nadeau et al., 2016; Neeter et al., 2003; Nichols, 1989; De Mesquita et al., 2018; O’Neill et al., 2019; Oloff et al., 2015; Ooi et al., 2015; Paavola et al., 2002; Paavola et al., 2002; Paavola et al., 2000; Paoloni et al., 2004; Papa, 2012; Pedowitz & Beck, 2017; Petersen, Welp & Rosenbaum, 2007; Pingel et al., 2013; Post et al., 2020; Praet et al., 2018; Rabello et al., 2020; Rasmussen et al., 2008; Reid et al., 2012; Reiter et al., 2004; Romero-Morales et al., 2019a; Rompe, Furia & Maffulli, 2009; Rompe et al., 2008; Rompe et al., 2007; Roos et al., 2004; Ryan et al., 2009; Saini et al., 2015; Santamato et al., 2019; Sayana & Maffulli, 2007; Scholes et al., 2018; Scott, Huisman & Khan, 2011; Sengkerij et al., 2009; Sharma & Maffulli, 2006; Silbernagel et al., 2007; Silbernagel et al., 2001; Simpson & Howard, 2009; Solomons et al., 2020; Sorosky et al., 2004; Stenson et al., 2018; Stergioulas et al., 2008; Syvertson et al., 2017; Tan & Chan, 2008; Thomas et al., 2010; Turner et al., 2020; Vallance et al., 2020; Van der Vlist et al., 2020; Van der Vlist et al., 2020; Van Sterkenburg et al., 2011; Verrall, Schofield & Brustad, 2011; Von Wehren et al., 2019; Wang et al., 2012; Wei et al., 2017; Welsh & Clodman, 1980; Xu et al., 2019; Zellers et al., 2019; Zhang et al., 2017; Zhang et al., 2020; Zhuang et al., 2019; Romero-Morales et al., 2019b) were included in this scoping review.

Figure 2 Preferred Reporting Items for Systematic Reviews and Meta-analysis flow diagram.

Characteristics of sources of evidence

In grouping the included articles by publication type, narrative reviews were the most common (27.2%) followed by cohort studies (19.6%), case control studies (18.8%), randomised controlled trials (12.7%), cross-sectional studies (10.8%), case reports (3.8%), protocols (3.2%), systematic reviews (1.9%), clinical guidelines (1.9%) and one consensus statement (0.6%). The years of publication of included studies ranged from 1980 to 2021, with 2017 to 2020 producing the most publications. Table 3 provides the general characteristics of the reviewed studies, including year of publication, type of publication, terminology and tendinopathy location.

Table 3 Characteristics of included studies.

Characteristics	No. of studies (n)	References	
Year of publication			
Before 1990	2	(Nichols, 1989; Welsh & Clodman, 1980)	
1990–1999	3	(Benazzo, Todesca & Ceciliani, 1997; Chazan, 1998; Fredericson, 1996)	
2000–2009	59	(Aldridge, 2004; Alfredson, 2003; Alfredson & Cook, 2007; Aronow, 2005; Azevedo et al., 2009; Bains & Porter, 2006; Barge-Caballero et al., 2008; Bjordal, Lopes-Martins & Iversen, 2006; Brown et al., 2006; Chester et al., 2008; Cook, Khan & Purdam, 2002; Courville, Coe & Hecht, 2009; Den Hartog, 2009; Furia & Rompe, 2007; Holmes & Lin, 2006; Hu & Flemister, 2008; Järvinen et al., 2001; Kader et al., 2002; Karjalainen et al., 2000; Khan et al., 2003; Knobloch, 2007; Knobloch et al., 2007; Knobloch et al., 2008; Lakshmanan & O’Doherty, 2004; Leung & Griffith, 2008; Lohrer & Nauck, 2009; Longo et al., 2009; Longo, Ronga & Maffulli, 2009; Maffulli & Kader, 2002; Maffulli et al., 2003; Maffulli et al., 2008; Maffulli, Sharma & Luscombe, 2004; Maffulli et al., 2008; Mafi, Lorentzon & Alfredson, 2001; Magnussen, Dunn & Thomson, 2009; Mayer et al., 2007; McShane, Ostick & McCabe, 2007; Neeter et al., 2003; Paavola et al., 2002; Paavola et al., 2002; Paavola et al., 2000; Paoloni et al., 2004; Petersen, Welp & Rosenbaum, 2007; Rasmussen et al., 2008; Reiter et al., 2004; Rompe, Furia & Maffulli, 2009; Rompe et al., 2008;Rompe et al., 2007; Roos et al., 2004; Ryan et al., 2009; Sayana & Maffulli, 2007; Sengkerij et al., 2009; Sharma & Maffulli, 2006; Silbernagel et al., 2007; Silbernagel et al., 2001; Simpson & Howard, 2009; Sorosky et al., 2004; Stergioulas et al., 2008; Tan & Chan, 2008)	
2010–2019	79	(Hutchison et al., 2013; Reiman et al., 2014; Abate & Salini, 2019; Alfredson & Spang, 2018; Asplund & Best, 2013; Barker-Davies et al., 2017; Baskerville et al., 2018; Benito, 2016; Bhatty, Khan & Zubairy, 2019; Boesen et al., 2017; Borda & Selhorst, 2017; Carcia et al., 2010; Cassel et al., 2018; Cheng, Zhang & Cai, 2016; Chimenti et al., 2016; Chimenti et al., 2017; Coombes et al., 2018; Creaby et al., 2017; Crill, Berlet & Hyer, 2014; De Jonge et al., 2011; De Marchi et al., 2018; Divani et al., 2010; Docking et al., 2015; Duthon et al., 2011; Ebbesen et al., 2018; Eckenrode, Kietrys & Stackhouse, 2019; Feilmeier, 2017; Finnamore et al., 2019; Florit et al., 2019; Gärdin et al., 2016; Habets et al., 2017; Hernández-Sánchez et al., 2018; Horn & McCollum, 2015; Hutchison et al., 2011; Irwin, 2010; Jayaseelan, Weber & Jonely, 2019; Jewson et al., 2017; Jowett, Richmond & Bedi, 2018; Kragsnaes et al., 2014; Krogh et al., 2016; Maffulli, Giai Via & Oliva, 2015; Maffulli, Giuseppe Longo & Denaro, 2012; Maffulli et al., 2011; Maffulli et al., 2012; Maffulli et al., 2019; Maffulli, Via & Oliva, 2014; Mansur et al., 2019; Mansur et al., 2017; Martin et al., 2018; McCormack et al., 2015; Murawski et al., 2014; Nadeau et al., 2016; De Mesquita et al., 2018; O’Neill et al., 2019; Oloff et al., 2015; Ooi et al., 2015; Papa, 2012; Pedowitz & Beck, 2017; Pingel et al., 2013; Praet et al., 2018; Reid et al., 2012; Romero-Morales et al., 2019a; Saini et al., 2015; Santamato et al., 2019; Scholes et al., 2018; Scott, Huisman & Khan, 2011; Stenson et al., 2018; Syvertson et al., 2017; Thomas et al., 2010; Van Sterkenburg et al., 2011; Verrall, Schofield & Brustad, 2011; Von Wehren et al., 2019; Wang et al., 2012; Wei et al., 2017; Xu et al., 2019; Zellers et al., 2019; Zhang et al., 2017; Zhuang et al., 2019; Romero-Morales et al., 2019b)	
2020–2021	16	(Millar et al., 2021; Aiyegbusi, Tella & Sanusi, 2020; Chimenti et al., 2020; Gatz et al., 2020; Hasani et al., 2020; Jukes, Scott & Solan, 2020; Maffulli et al., 2020; Mantovani et al., 2020; Post et al., 2020; Rabello et al., 2020; Solomons et al., 2020; Turner et al., 2020; Vallance et al., 2020; Van der Vlist et al., 2020; Van der Vlist et al., 2020; Zhang et al., 2020)	
Type of publication			
Clinical guidelines	3	(Carcia et al., 2010; Martin et al., 2018; Thomas et al., 2010)	
Consensus statement	1	(Xu et al., 2019)	
Systematic reviews	3	(Reiman et al., 2014; Hutchison et al., 2011; Magnussen, Dunn & Thomson, 2009)	
RCT	20	(Bjordal, Lopes-Martins & Iversen, 2006; Boesen et al., 2017; Brown et al., 2006; Ebbesen et al., 2018; Gatz et al., 2020; Knobloch et al., 2007; Krogh et al., 2016; Mafi, Lorentzon & Alfredson, 2001; Mayer et al., 2007; Paoloni et al., 2004; Petersen, Welp & Rosenbaum, 2007; Rasmussen et al., 2008; Rompe, Furia & Maffulli, 2009; Rompe et al., 2008; Rompe et al., 2007; Roos et al., 2004; Silbernagel et al., 2001; Solomons et al., 2020; Stergioulas et al., 2008; Van der Vlist et al., 2020)	
Cohort studies	31	(Alfredson & Spang, 2018; Barge-Caballero et al., 2008; Cheng, Zhang & Cai, 2016; Chester et al., 2008; Crill, Berlet & Hyer, 2014; Duthon et al., 2011; Florit et al., 2019; Jowett, Richmond & Bedi, 2018; Karjalainen et al., 2000; Khan et al., 2003; Knobloch, 2007; Knobloch et al., 2008; Maffulli et al., 2008; Mansur et al., 2019; McCormack et al., 2015; Murawski et al., 2014; O’Neill et al., 2019; Oloff et al., 2015; Paavola et al., 2002; Paavola et al., 2000; Sayana & Maffulli, 2007; Silbernagel et al., 2007; Stenson et al., 2018; Syvertson et al., 2017; Von Wehren et al., 2019; Wei et al., 2017; Welsh & Clodman, 1980; Zellers et al., 2019; Zhang et al., 2020; Zhuang et al., 2019)	
Case control studies	30	(Hutchison et al., 2013; Abate & Salini, 2019; Azevedo et al., 2009; Cassel et al., 2018; Chimenti et al., 2016; Chimenti et al., 2020; Coombes et al., 2018; Creaby et al., 2017; Eckenrode, Kietrys & Stackhouse, 2019; Gärdin et al., 2016; Hernández-Sánchez et al., 2018; Holmes & Lin, 2006; Jewson et al., 2017; Leung & Griffith, 2008; Lohrer & Nauck, 2009; Maffulli et al., 2003; Nadeau et al., 2016; Neeter et al., 2003; De Mesquita et al., 2018; Ooi et al., 2015; Pingel et al., 2013; Rabello et al., 2020; Reid et al., 2012; Reiter et al., 2004; Romero-Morales et al., 2019a; Ryan et al., 2009; Sengkerij et al., 2009; Verrall, Schofield & Brustad, 2011; Zhang et al., 2017; Romero-Morales et al., 2019b)	
Cross-sectional studies	17	(Aiyegbusi, Tella & Sanusi, 2020; De Jonge et al., 2011; De Marchi et al., 2018; Divani et al., 2010; Docking et al., 2015; Finnamore et al., 2019; Kragsnaes et al., 2014; Longo et al., 2009; Maffulli et al., 2008; Mantovani et al., 2020; Praet et al., 2018; Santamato et al., 2019; Scholes et al., 2018; Turner et al., 2020; Vallance et al., 2020; Van der Vlist et al., 2020; Wang et al., 2012)	
Narrative reviews	43	(Millar et al., 2021; Aldridge, 2004; Alfredson, 2003; Alfredson & Cook, 2007; Aronow, 2005; Asplund & Best, 2013; Bains & Porter, 2006; Baskerville et al., 2018; Benazzo, Todesca & Ceciliani, 1997; Bhatty, Khan & Zubairy, 2019; Chazan, 1998; Chimenti et al., 2017; Cook, Khan & Purdam, 2002; Courville, Coe & Hecht, 2009; Den Hartog, 2009; Feilmeier, 2017; Fredericson, 1996; Furia & Rompe, 2007; Horn & McCollum, 2015; Hu & Flemister, 2008; Irwin, 2010; Järvinen et al., 2001; Jukes, Scott & Solan, 2020; Kader et al., 2002; Longo, Ronga & Maffulli, 2009; Maffulli, Giai Via & Oliva, 2015; Maffulli, Giuseppe Longo & Denaro, 2012; Maffulli & Kader, 2002; Maffulli et al., 2020; Maffulli et al., 2012; Maffulli et al., 2019; Maffulli, Sharma & Luscombe, 2004; Maffulli, Via & Oliva, 2014; McShane, Ostick & McCabe, 2007; Nichols, 1989; Paavola et al., 2002; Pedowitz & Beck, 2017; Saini et al., 2015; Scott, Huisman & Khan, 2011; Sharma & Maffulli, 2006; Simpson & Howard, 2009; Sorosky et al., 2004; Tan & Chan, 2008)	
Case reports	6	(Benito, 2016; Borda & Selhorst, 2017; Jayaseelan, Weber & Jonely, 2019; Maffulli et al., 2011; Papa, 2012; Van Sterkenburg et al., 2011)	
Protocols	5	(Barker-Davies et al., 2017; Habets et al., 2017; Hasani et al., 2020; Mansur et al., 2017; Post et al., 2020)	
Terminology			
Tendon pain	2	(Neeter et al., 2003; Silbernagel et al., 2001)	
Tendinitis	3	(Aldridge, 2004; Paavola et al., 2000; Welsh & Clodman, 1980)	
Tendinosis	3	(Gärdin et al., 2016; Karjalainen et al., 2000; Wei et al., 2017)	
Tendinopathy	144	(Millar et al., 2021; Hutchison et al., 2013; Reiman et al., 2014; Abate & Salini, 2019; Aiyegbusi, Tella & Sanusi, 2020; Alfredson, 2003; Alfredson & Cook, 2007; Alfredson & Spang, 2018; Aronow, 2005; Asplund & Best, 2013; Azevedo et al., 2009; Bains & Porter, 2006; Barge-Caballero et al., 2008; Barker-Davies et al., 2017; Baskerville et al., 2018; Benazzo, Todesca & Ceciliani, 1997; Benito, 2016; Bhatty, Khan & Zubairy, 2019; Bjordal, Lopes-Martins & Iversen, 2006; Boesen et al., 2017; Borda & Selhorst, 2017; Brown et al., 2006; Carcia et al., 2010; Cassel et al., 2018; Cheng, Zhang & Cai, 2016; Chester et al., 2008; Chimenti et al., 2016; Chimenti et al., 2017; Chimenti et al., 2020; Cook, Khan & Purdam, 2002; Coombes et al., 2018; Courville, Coe & Hecht, 2009; Creaby et al., 2017; Crill, Berlet & Hyer, 2014; De Jonge et al., 2011; De Marchi et al., 2018; Divani et al., 2010; Docking et al., 2015; Duthon et al., 2011; Ebbesen et al., 2018; Eckenrode, Kietrys & Stackhouse, 2019; Feilmeier, 2017; Finnamore et al., 2019; Florit et al., 2019; Furia & Rompe, 2007; Gatz et al., 2020; Habets et al., 2017; Hasani et al., 2020; Hernández-Sánchez et al., 2018; Holmes & Lin, 2006; Horn & McCollum, 2015; Hu & Flemister, 2008; Hutchison et al., 2011; Irwin, 2010; Järvinen et al., 2001; Jayaseelan, Weber & Jonely, 2019; Jewson et al., 2017; Jowett, Richmond & Bedi, 2018; Jukes, Scott & Solan, 2020; Kader et al., 2002; Khan et al., 2003; Knobloch, 2007; Knobloch et al., 2007; Knobloch et al., 2008; Kragsnaes et al., 2014; Krogh et al., 2016; Lakshmanan & O’Doherty, 2004; Leung & Griffith, 2008; Lohrer & Nauck, 2009; Longo et al., 2009; Longo, Ronga & Maffulli, 2009; Maffulli, Giai Via & Oliva, 2015; Maffulli, Giuseppe Longo & Denaro, 2012; Maffulli & Kader, 2002; Maffulli et al., 2003; Maffulli et al., 2011; Maffulli et al., 2020; Maffulli et al., 2012; Maffulli et al., 2008; Maffulli et al., 2019; Maffulli, Sharma & Luscombe, 2004; Maffulli, Via & Oliva, 2014; Maffulli et al., 2008; Mafi, Lorentzon & Alfredson, 2001; Magnussen, Dunn & Thomson, 2009; Mansur et al., 2019; Mansur et al., 2017; Mantovani et al., 2020; Martin et al., 2018; Mayer et al., 2007; McCormack et al., 2015; McShane, Ostick & McCabe, 2007; Murawski et al., 2014; Nadeau et al., 2016; De Mesquita et al., 2018; O’Neill et al., 2019; Oloff et al., 2015; Ooi et al., 2015; Paavola et al., 2002; Paavola et al., 2002; Paoloni et al., 2004; Papa, 2012; Petersen, Welp & Rosenbaum, 2007; Pingel et al., 2013; Post et al., 2020; Praet et al., 2018; Rabello et al., 2020; Rasmussen et al., 2008; Reid et al., 2012; Reiter et al., 2004; Romero-Morales et al., 2019a; Rompe, Furia & Maffulli, 2009; Rompe et al., 2008; Rompe et al., 2007; Roos et al., 2004; Ryan et al., 2009; Saini et al., 2015; Santamato et al., 2019; Sayana & Maffulli, 2007; Scholes et al., 2018; Sengkerij et al., 2009; Sharma & Maffulli, 2006; Silbernagel et al., 2007; Simpson & Howard, 2009; Solomons et al., 2020; Sorosky et al., 2004; Stenson et al., 2018; Stergioulas et al., 2008; Syvertson et al., 2017; Tan & Chan, 2008; Thomas et al., 2010; Turner et al., 2020; Vallance et al., 2020; Van der Vlist et al., 2020; Van der Vlist et al., 2020; Van Sterkenburg et al., 2011; Verrall, Schofield & Brustad, 2011; Von Wehren et al., 2019; Xu et al., 2019; Zellers et al., 2019; Zhang et al., 2017; Zhang et al., 2020; Zhuang et al., 2019; Romero-Morales et al., 2019b)	
Combined terminology	7	(Chazan, 1998; Den Hartog, 2009; Fredericson, 1996; Nichols, 1989; Pedowitz & Beck, 2017; Scott, Huisman & Khan, 2011; Wang et al., 2012)	
Tendinopathy location			
Insertional	21	(Aldridge, 2004; Benazzo, Todesca & Ceciliani, 1997; Benito, 2016; Cheng, Zhang & Cai, 2016; Chimenti et al., 2016; Chimenti et al., 2017; Den Hartog, 2009; Hu & Flemister, 2008; Irwin, 2010; Maffulli et al., 2019; Mansur et al., 2019; Mansur et al., 2017; McCormack et al., 2015; Rompe et al., 2008; Stenson et al., 2018; Thomas et al., 2010; Wei et al., 2017; Xu et al., 2019; Zellers et al., 2019; Zhang et al., 2020; Zhuang et al., 2019)	
Midportion	83	(Hutchison et al., 2013; Reiman et al., 2014; Abate & Salini, 2019; Alfredson, 2003; Alfredson & Cook, 2007; Azevedo et al., 2009; Bains & Porter, 2006; Boesen et al., 2017; Borda & Selhorst, 2017; Brown et al., 2006; Carcia et al., 2010; Chester et al., 2008; Courville, Coe & Hecht, 2009; Creaby et al., 2017; Crill, Berlet & Hyer, 2014; De Jonge et al., 2011; De Marchi et al., 2018; Divani et al., 2010; Duthon et al., 2011; Feilmeier, 2017; Finnamore et al., 2019; Gärdin et al., 2016; Habets et al., 2017; Hasani et al., 2020; Hutchison et al., 2011; Järvinen et al., 2001; Jayaseelan, Weber & Jonely, 2019; Jowett, Richmond & Bedi, 2018; Kader et al., 2002; Knobloch et al., 2008; Krogh et al., 2016; Lakshmanan & O’Doherty, 2004; Lohrer & Nauck, 2009; Longo, Ronga & Maffulli, 2009; Maffulli, Giuseppe Longo & Denaro, 2012; Maffulli & Kader, 2002; Maffulli et al., 2003; Maffulli et al., 2011; Maffulli et al., 2008; Maffulli, Sharma & Luscombe, 2004; Maffulli, Via & Oliva, 2014; Maffulli et al., 2008; Mafi, Lorentzon & Alfredson, 2001; Magnussen, Dunn & Thomson, 2009; Martin et al., 2018; Mayer et al., 2007; McShane, Ostick & McCabe, 2007; Murawski et al., 2014; Nadeau et al., 2016; Neeter et al., 2003; O’Neill et al., 2019; Paavola et al., 2002; Paavola et al., 2002; Paavola et al., 2000; Paoloni et al., 2004; Papa, 2012; Petersen, Welp & Rosenbaum, 2007; Pingel et al., 2013; Praet et al., 2018; Reid et al., 2012; Romero-Morales et al., 2019a; Rompe, Furia & Maffulli, 2009; Rompe et al., 2007; Roos et al., 2004; Ryan et al., 2009; Santamato et al., 2019; Sayana & Maffulli, 2007; Scholes et al., 2018; Scott, Huisman & Khan, 2011; Sengkerij et al., 2009; Sharma & Maffulli, 2006; Silbernagel et al., 2007; Simpson & Howard, 2009; Solomons et al., 2020; Stergioulas et al., 2008; Syvertson et al., 2017; Tan & Chan, 2008; Van der Vlist et al., 2020; Van der Vlist et al., 2020; Van Sterkenburg et al., 2011; Von Wehren et al., 2019; Wang et al., 2012; Romero-Morales et al., 2019b)	
Both	35	(Alfredson & Spang, 2018; Aronow, 2005; Asplund & Best, 2013; Barker-Davies et al., 2017; Bhatty, Khan & Zubairy, 2019; Chazan, 1998; Chimenti et al., 2020; Cook, Khan & Purdam, 2002; Coombes et al., 2018; Docking et al., 2015; Eckenrode, Kietrys & Stackhouse, 2019; Fredericson, 1996; Furia & Rompe, 2007; Gatz et al., 2020; Hernández-Sánchez et al., 2018; Horn & McCollum, 2015; Jukes, Scott & Solan, 2020; Karjalainen et al., 2000; Khan et al., 2003; Knobloch, 2007; Knobloch et al., 2007; Kragsnaes et al., 2014; Maffulli, Giai Via & Oliva, 2015; Maffulli et al., 2020; Ooi et al., 2015; Pedowitz & Beck, 2017; Post et al., 2020; Rabello et al., 2020; Reiter et al., 2004; Saini et al., 2015; Turner et al., 2020; Vallance et al., 2020; Verrall, Schofield & Brustad, 2011; Welsh & Clodman, 1980; Zhang et al., 2017)	
Not specified	20	(Millar et al., 2021; Aiyegbusi, Tella & Sanusi, 2020; Barge-Caballero et al., 2008; Baskerville et al., 2018; Bjordal, Lopes-Martins & Iversen, 2006; Cassel et al., 2018; Ebbesen et al., 2018; Florit et al., 2019; Holmes & Lin, 2006; Jewson et al., 2017; Leung & Griffith, 2008; Longo et al., 2009; Maffulli et al., 2012; Mantovani et al., 2020; Nichols, 1989; De Mesquita et al., 2018; Oloff et al., 2015; Rasmussen et al., 2008; Silbernagel et al., 2001; Sorosky et al., 2004)	
Note:

n, number; RCT, randomised controlled trial.

As highlighted in Fig. 3, the terminology used to describe tendon pain varied, with ‘tendinopathy’ being the most prevalent term used to describe tendon pain. Thus, during this scoping review, tendinopathy, will be used to describe pain located in the Achilles tendon that impairs function.

Figure 3 Terminology used to describe the clinical presentation of Achilles tendon pain and impaired function.

Results of individual sources of evidence

Clinical guidelines and consensus statements

Two of the included clinical guidelines (Carcia et al., 2010; Martin et al., 2018) discussed midportion Achilles tendinopathy, with one clinical guideline (Thomas et al., 2010) and one consensus statement (Xu et al., 2019) discussing insertional Achilles tendinopathy (Table 4). Clinical measures used to diagnose Achilles tendinopathy was consistent across the clinical guidelines and consensus statement, with location of pain being the main differentiating factor between diagnosing midportion or insertional tendinopathy. Common methods with which midportion tendinopathy was diagnosed included subjective reporting of pain located in the Achilles tendon 2–6 cm above the calcaneal insertion that is increased with tendon loading and reported tendon stiffness. Similarly, insertional tendinopathy was diagnosed via subjective reporting of pain and swelling at the calcaneal insertion of the Achilles tendon. Pain on palpation was utilised to confirm clinical diagnosis in both midportion and insertional tendinopathy. While additional objective tests for midportion tendinopathy included the ‘Painful Arc Sign’ and ‘Royal London Hospital Test’.

Table 4 Clinical guidelines and consensus statement.

Author	Year	Location	Subjective history	Clinical tests	
Carcia et al. (2010)	2010	Midportion	Location of pain (2–6 cm above calcaneal insertion)
Pain with tendon loading
Tendon stiffness	Pain on palpation
Painful Arc Sign
Royal London Hospital Test
Single-leg heel Raise
Hopping	
Thomas et al. (2010)	2010	Insertional	Location of pain (insertion)
Pain with tendon loading
Swelling	Pain on palpation
Localised tendon thickening on palpation	
Martin et al. (2018)	2018	Midportion	Location of pain (2–6 cm above calcaneal insertion)
Pain with tendon loading
Tendon stiffness	Pain on palpation
Painful Arc Sign
Royal London Hospital Test	
Xu et al. (2019)	2019	Insertional	Location of pain (insertion)
Pain with tendon loading	Pain on palpation
Localised swelling on palpation
Pain with active dorsiflexion
Silverskiold Test	
Note:

cm, centimetres.

Systematic reviews

All three included systematic reviews assessed midportion Achilles tendinopathy (Table 5) (Reiman et al., 2014; Hutchison et al., 2011; Magnussen, Dunn & Thomson, 2009). Subjective reporting of pain with tendon loading was included as a diagnostic feature of midportion Achilles tendinopathy in all three systematic reviews (Leung & Griffith, 2008; Hutchison et al., 2011; Magnussen, Dunn & Thomson, 2009). Two of the systematic reviews (Reiman et al., 2014; Hutchison et al., 2011) identified the location of tendon pain as 2–6 cm above the calcaneal insertion, with one (Magnussen, Dunn & Thomson, 2009) defining the location of tendon pain as 2–7 cm above the calcaneal insertion. Palpation of the Achilles tendon, passive dorsiflexion, pain with single-leg heel raise and pain hopping or jumping were included as clinical tests in all included systematic reviews (Reiman et al., 2014; Hutchison et al., 2011; Magnussen, Dunn & Thomson, 2009). Two of the systematic reviews (Reiman et al., 2014; Hutchison et al., 2011) included the ‘Painful Arc Sign’ and ‘Royal London Hospital Test’ as diagnostic measures for midportion Achilles tendinopathy.

Table 5 Systematic reviews.

Author	Year	Sample size	Location	Subjective history	Clinical tests	
Magnussen, Dunn & Thomson (2009)	2009	677 (M/F = 347/330)	Midportion	Location of pain (2–7 cm above calcaneal insertion)
Pain with tendon loading	Pain on palpation
Localised swelling on palpation
Localised tendon thickening on palpation
Pain with passive dorsiflexion
Single-leg heel raise
Hopping	
Hutchison et al. (2011)	2011	578 (M/F = not specified)	Midportion	Location of pain (2–6 cm above calcaneal insertion)
Pain with tendon loading
Tendon stiffness	Pain on palpation
Localised swelling on palpation
Localised tendon thickening on palpation
Painful Arc Sign
Royal London Hospital Test
Reduced dorsiflexion
Single-leg heel Raise
Jump test	
Reiman et al. (2014)	2014	31 (M/F = 27/4)	Midportion	Location of pain (2–6 cm above calcaneal insertion)
Pain with tendon loading
Tendon stiffness	Pain on palpation
Localised swelling on palpation
Localised tendon thickening on palpation
Painful Arc Sign
Royal London Hospital Test
Pain with dorsiflexion
Single-leg heel Raise
Hopping	
Note:

cm, centimetres; M, male; F, female.

Randomised controlled trials

Table 6 highlights the characteristics of the included randomised controlled trials. Thirteen of the included studies (Boesen et al., 2017; Brown et al., 2006; Krogh et al., 2016; Mafi, Lorentzon & Alfredson, 2001; Mayer et al., 2007; Paoloni et al., 2004; Petersen, Welp & Rosenbaum, 2007; Rompe, Furia & Maffulli, 2009; Rompe et al., 2007; Roos et al., 2004; Solomons et al., 2020; Stergioulas et al., 2008; Van der Vlist et al., 2020) investigated midportion Achilles tendinopathy, one study (Rompe et al., 2008) investigated insertional Achilles tendinopathy, two studies (Gatz et al., 2020; Knobloch et al., 2007) investigated both insertional and midportion Achilles tendinopathy, and four studies (Bjordal, Lopes-Martins & Iversen, 2006; Ebbesen et al., 2018; Rasmussen et al., 2008; Silbernagel et al., 2001) did not specify a location of interest. All of the included randomised controlled trials used location of pain as a diagnostic feature of Achilles tendinopathy. Eight of the studies (Knobloch et al., 2007; Mafi, Lorentzon & Alfredson, 2001; Paoloni et al., 2004; Petersen, Welp & Rosenbaum, 2007; Rompe, Furia & Maffulli, 2009; Rompe et al., 2008; Roos et al., 2004; Stergioulas et al., 2008) which assessed midportion tendinopathy defined the location of pain as 2–6 cm above the calcaneal insertion, with three studies (Boesen et al., 2017; Krogh et al., 2016; Van der Vlist et al., 2020) defining midportion tendinopathy as 2–7 cm above the calcaneal insertion. Of the included studies, 17 included symptom duration as part of their diagnostic criteria, with various durations including four weeks (Roos et al., 2004), six weeks (Brown et al., 2006), two months (Gatz et al., 2020; Van der Vlist et al., 2020), three months (Boesen et al., 2017; Ebbesen et al., 2018; Mafi, Lorentzon & Alfredson, 2001; Paoloni et al., 2004; Petersen, Welp & Rosenbaum, 2007; Rasmussen et al., 2008; Silbernagel et al., 2001; Solomons et al., 2020) and six months (Mayer et al., 2007; Rompe, Furia & Maffulli, 2009; Rompe et al., 2008; Rompe et al., 2007; Stergioulas et al., 2008). Palpation was the most commonly used objective test, with 15 of the included studies (Bjordal, Lopes-Martins & Iversen, 2006; Boesen et al., 2017; Brown et al., 2006; Gatz et al., 2020; Krogh et al., 2016; Mafi, Lorentzon & Alfredson, 2001; Mayer et al., 2007; Paoloni et al., 2004; Petersen, Welp & Rosenbaum, 2007; Rasmussen et al., 2008; Rompe et al., 2008; Roos et al., 2004; Silbernagel et al., 2001; Stergioulas et al., 2008; Van der Vlist et al., 2020) using palpation to assess pain, localised tendon thickening or localised swelling. Four studies (Ebbesen et al., 2018; Knobloch et al., 2007; Rompe, Furia & Maffulli, 2009; Rompe et al., 2007) used solely subjective history to diagnose Achilles tendinopathy.

Table 6 Randomised controlled trials.

Author	Year	Sample size	Location	Subjective history	Clinical tests	
Mafi, Lorentzon & Alfredson (2001)	2001	44 (M/F = 24/20)	Midportion	Location of pain (2–6 cm above calcaneal insertion)
Duration of symptoms (>3 months)	Pain on palpation	
Silbernagel et al. (2001)	2001	49 (M/F = 36/13)	Not specified	Location of pain
Duration of symptoms (>3 months)	Pain on palpation
Single leg heel raise
Hopping
Range of motion	
Paoloni et al. (2004)	2004	65 (M/F = 40/25)	Midportion	Location of pain (2–6 cm above calcaneal insertion)
Duration of symptoms (>3 months)
Gradual onset of pain	Pain on palpation
Localised tendon thickening on palpation
Hopping	
Roos et al. (2004)	2004	44 (M/F = 21/23)	Midportion	Location of pain (2–6 cm above calcaneal insertion)
Duration of symptoms (>4 weeks)
Pain with tendon loading	Pain on palpation	
Bjordal, Lopes-Martins & Iversen (2006)	2006	7 (M/F = not specified)	Not specified	Location of pain
Pain with tendon loading	Pain on palpation
Hopping	
Brown et al. (2006)	2006	26 (M/F = 17/9)	Midportion	Location of pain
Duration of symptoms (>6 weeks)
Gradual onset of pain
Pain with tendon loading	Pain on palpation
Single-leg heel raise
Hopping	
Knobloch et al. (2007)	2007	20 (M/F = 11/9)	Insertional Midportion	Location of pain (2–6 cm above calcaneal insertion)
Location of pain (insertion)
Pain with tendon loading
Swelling	Not specified	
Mayer et al. (2007)	2007	31 (M/F = 31/0)	Midportion	Location of pain
Duration of symptoms (>6 months)
Pain with tendon loading	Pain on palpation
Localised tendon thickening on palpation	
Petersen, Welp & Rosenbaum (2007)	2007	100 (M/F = 60/40)	Midportion	Location of pain (2–6 cm above calcaneal insertion)
Duration of symptoms (>3 months)
Gradual onset of pain
Pain with tendon loading	Pain on palpation
Localised tendon thickening on palpation	
Rompe et al. (2007)	2007	75 (M/F = 29/46)	Midportion	Location of pain (2–6 cm above calcaneal insertion)
Duration of symptoms (>6 months)
Pain with tendon loading
Swelling	Not specified	
Rasmussen et al. (2008)	2008	48 (M/F = 28/20)	Not specified	Location of pain
Duration of symptoms (>3 months)	Pain on palpation
Localised swelling on palpation
Pain with dorsiflexion	
Rompe et al. (2008)	2008	50 (M/F = 20/30)	Insertional	Location of pain
Pain with tendon loading
Duration of symptoms (>6 months)	Pain on palpation
Painful Arc Sign
Royal London Hospital Test	
Stergioulas et al. (2008)	2008	40 (M/F = 25/15)	Midportion	Location of pain (2–6 cm above calcaneal insertion)
Duration of symptoms (>6 months)
Pain with tendon loading	Pain on palpation
Reduced active dorsiflexion	
Rompe, Furia & Maffulli (2009)	2009	68 (M/F = 30/38)	Midportion	Location of pain (2–6 cm above calcaneal insertion)
Duration of symptoms (>6 months)
Pain with tendon loading
Swelling	Not specified	
Krogh et al. (2016)	2016	24 (M/F = 13/11)	Midportion	Location of pain (2–7 cm above calcaneal insertion)
Pain with tendon loading	Pain on palpation
Localised tendon thickening on palpation	
Boesen et al. (2017)	2017	60 (M/F = 60/0)	Midportion	Location of pain (2–7 cm above calcaneal insertion)
Duration of symptoms (>3 months)	Pain on palpation
Localised tendon thickening on palpation
Single-leg heel raise	
Ebbesen et al. (2018)	2018	44 (M/F = 25/19)	Not specified	Location of pain
Duration of symptoms (>3 months)	Not specified	
Gatz et al. (2020)	2020	42 (M/F = 20/22)	Insertional Midportion	Pain with tendon loading
Duration of symptoms (>2 months)
Tendon stiffness	Pain on palpation	
Solomons et al. (2020)	2020	52 (M/F = 24/28)	Midportion	Location of pain
Duration of symptoms (>3 months)
Pain with tendon loading	Double leg heel raise
Single leg heel raise
Jump
Hopping	
Van der Vlist et al. (2020)	2020	91 (M/F = 45/46)	Midportion	Location of pain (2–7 cm above calcaneal insertion)
Duration of symptoms (>2 months)
Pain with tendon loading	Pain on palpation
Localised swelling on palpation	
Note:

cm, centimetres; M, male; F, female.

Cohort studies

Of the included cohort studies, 21 were prospective cohort studies (Cheng, Zhang & Cai, 2016; Chester et al., 2008; Crill, Berlet & Hyer, 2014; Duthon et al., 2011; Jowett, Richmond & Bedi, 2018; Karjalainen et al., 2000; Khan et al., 2003; Knobloch, 2007; Knobloch et al., 2008; Maffulli et al., 2008; Mansur et al., 2019; McCormack et al., 2015; O’Neill et al., 2019; Oloff et al., 2015; Paavola et al., 2002; Paavola et al., 2000; Sayana & Maffulli, 2007; Silbernagel et al., 2007; Syvertson et al., 2017; Zhuang et al., 2019) and 10 were retrospective cohort studies (Alfredson & Spang, 2018; Barge-Caballero et al., 2008; Florit et al., 2019; Murawski et al., 2014; Stenson et al., 2018; Von Wehren et al., 2019; Wei et al., 2017; Welsh & Clodman, 1980; Zellers et al., 2019; Zhang et al., 2020). Midportion Achilles tendinopathy was investigated in 15 studies (Chester et al., 2008; Crill, Berlet & Hyer, 2014; Duthon et al., 2011; Jowett, Richmond & Bedi, 2018; Knobloch et al., 2008; Lakshmanan & O’Doherty, 2004; Maffulli et al., 2008; Murawski et al., 2014; O’Neill et al., 2019; Paavola et al., 2002; Paavola et al., 2000; Sayana & Maffulli, 2007; Silbernagel et al., 2007; Syvertson et al., 2017; Von Wehren et al., 2019), insertional tendinopathy was investigated in eight studies (Cheng, Zhang & Cai, 2016; Mansur et al., 2019; McCormack et al., 2015; Stenson et al., 2018; Wei et al., 2017; Zellers et al., 2019; Zhang et al., 2020; Zhuang et al., 2019), both insertional and midportion tendinopathy was investigated in five studies (Alfredson & Spang, 2018; Karjalainen et al., 2000; Khan et al., 2003; Knobloch, 2007; Welsh & Clodman, 1980), and three studies did not specify tendinopathy location (Table 7) (Barge-Caballero et al., 2008; Florit et al., 2019; Oloff et al., 2015).

Table 7 Cohort studies.

Author	Year	Study design	Sample size	Location	Subjective history	Clinical tests	
Welsh & Clodman (1980)	1980	Retrospective	50 (M/F = 28/22)	Insertional Midportion	Location of pain (2–4 cm above calcaneal insertion)
Pain with tendon loading	Pain on palpation	
Karjalainen et al. (2000)	2000	Prospective	100 (M/F = 75/25)	Insertional Midportion	Location of pain
Duration of symptoms (not specified)	Pain on palpation
Localised tendon thickening on palpation	
Paavola et al. (2000)	2000	Prospective	107 (M/F = 78/29)	Midportion	Pain with tendon loading
Duration of symptoms (<6 months)	Pain on palpation
Range of motion
Single-leg heel raise
Single-leg stance	
Paavola et al. (2002)	2002	Prospective	42 (M/F = 29/13)	Midportion	Pain with tendon loading	Pain on palpation
Localised tendon thickening on palpation
Localised swelling on palpation Range of Motion
Single-leg heel raise	
Khan et al. (2003)	2003	Prospective	45 (M/F = 27/18)	Insertional Midportion	Pain with tendon loading
Tendon stiffness
VISA-A	Pain on palpation	
Lakshmanan & O’Doherty (2004)	2004	Prospective	15 (M/F = 12/3)	Midportion	Location of pain
Duration of symptoms (>6 months)	Not specified	
Knobloch (2007)	2007	Prospective	64 (M/F = 39/25)	Insertional Midportion	Location of pain (2–6 cm above calcaneal insertion)-midportion
Location of pain (insertion)
Pain with tendon loading
Swelling	Not specified	
Sayana & Maffulli (2007)	2007	Prospective	34 (M/F = 18/16)	Midportion	Location of pain (2–6 cm above calcaneal insertion)
Pain with tendon loading	Pain on palpation
Painful Arc Sign
Royal London Hospital Test	
Silbernagel et al. (2007)	2007	Prospective	37 (M/F = 20/17)	Midportion	Location of pain
Duration of symptoms (>2 months)
Pain with tendon loading
Swelling	Counter movement Jump Hopping Heel raise	
Barge-Caballero et al. (2008)	2008	Retrospective	242 (M/F = 191/51)	Not specified	Pain with tendon loading	Pain on palpation	
Chester et al. (2008)	2008	Prospective	16 (M/F = 11/5)	Midportion	Location of pain (2–6 cm above calcaneal insertion)
Duration of symptoms (>3 months)	Pain on palpation
Localised swelling on palpation	
Knobloch et al. (2008)	2008	Prospective	121 (M/F = 74/47)	Midportion	Location of pain (2–6 cm above calcaneal insertion)
Duration of symptoms (>3 months)	Pain on palpation
Localised swelling on palpation	
Maffulli et al. (2008)	2008	Prospective	45 (M/F = 29/16)	Midportion	Location of pain (2–6 cm above calcaneal insertion)
Pain with tendon loading	Pain on palpation
Painful Arc Sign
Royal London Hospital Test	
Duthon et al. (2011)	2011	Prospective	14 (M/F = 11/3)	Midportion	Location of pain
Duration of symptoms (>1 year)	Localised tendon thickening on palpation
Range of Motion
Plantarflexion strength
Silfverskiold test	
Crill, Berlet & Hyer (2014)	2014	Prospective	25 (M/F = not specified)	Midportion	Location of pain (2–4 cm above calcaneal insertion)	Pain on palpation
Localised tendon thickening on palpation
Single-leg heel raise	
Murawski et al. (2014)	2014	Retrospective	32 (M/F = 21/11)	Midportion	Location of pain (2–7 cm above calcaneal insertion)
Pain with tendon loading
Tendon Stiffness	Pain on palpation
Localised tendon thickening on palpation	
McCormack et al. (2015)	2015	Prospective	15 (M/F = 4/11)	Insertional	Location of pain (distal 2 cm)
Duration of symptoms (>6 weeks)
Pain with tendon loading
VISA-A	Pain on palpation	
Oloff et al. (2015)	2015	Prospective	26 (M/F = not specified)	Not specified	Location of pain
Duration of symptoms (>6 months)	Not specified	
Cheng, Zhang & Cai (2016)	2016	Prospective	42 (M/F = 29/13)	Insertional	Location of pain (distal 2 cm)
Duration of symptoms (>6 months)
Swelling
Pain with tendon loading	Not specified	
Syvertson et al. (2017)	2017	Prospective	11 (M/F = 4/7)	Midportion	Location of pain (2–6 cm above calcaneal insertion)
Pain with tendon loading
Tendon stiffness	Pain on palpation	
Wei et al. (2017)	2017	Retrospective	68 (M/F = 53/15)	Insertional	Location of pain
Duration of symptoms (>6 months)
Pain with tendon loading	Pain on palpation	
Alfredson & Spang (2018)	2018	Retrospective	771 (M/F = 481/290)	Insertional Midportion	Pain with tendon loading
VISA-A	Not specified	
Jowett, Richmond & Bedi (2018)	2018	Prospective	26 (M/F = 13/13)	Midportion	Location of pain
Localised swelling
Duration of symptoms (>6 months)	Pain on palpation
Localised tendon thickening on palpation	
Stenson et al. (2018)	2018	Retrospective	664 (M/F = 312/352)	Insertional	Location of pain (insertion)
Duration of symptoms (>3 months)	Range of motion	
Florit et al. (2019)	2019	Retrospective	110 (M/F = 103/7)	Not specified	Location of pain
Pain with tendon loading	Pain on palpation
Pain with tendon loading tests (not specified)	
Mansur et al. (2019)	2019	Prospective	19 (M/F = 11/8)	Insertional	Location of pain (distal 2 cm)	Pain on palpation	
O’Neill et al. (2019)	2019	Prospective	16 (M/F = 11/5)	Midportion	Location of pain
Duration of symptoms (>3 months)
Pain with tendon loading	Pain on palpation
Painful Arc Sign
Royal London Hospital Test	
Von Wehren et al. (2019)	2019	Retrospective	50 (M/F = 27/23)	Midportion	Location of pain
Duration of symptoms (>6 weeks)
Pain with tendon loading	Pain on palpation
Localised swelling on palpation	
Zellers et al. (2019)	2019	Retrospective	56 (M/F = 25/31)	Insertional	Location of pain (insertion)	Pain on palpation	
Zhuang et al. (2019)	2019	Prospective	28 (M/F = 17/11)	Insertional	Location of pain (insertion)	Pain on palpation
Localised tendon thickening on palpation Pain with resisted plantarflexion Reduced plantarflexion strength Heel raise test	
Zhang et al. (2020)	2020	Retrospective	33 (M/F = 31/2)	Insertional	Location of pain (insertion)
Duration of symptoms (>3 months)	Not specified	
Note:

cm, centimetres; M, male; F, female; VISA-A, Victorian Institute of Sport Assessment-Achilles.

Location of pain was the most prominent diagnostic feature, with 26 studies (Cheng, Zhang & Cai, 2016; Chester et al., 2008; Crill, Berlet & Hyer, 2014; Duthon et al., 2011; Florit et al., 2019; Jowett, Richmond & Bedi, 2018; Karjalainen et al., 2000; Knobloch, 2007; Knobloch et al., 2008; Lakshmanan & O’Doherty, 2004; Maffulli et al., 2008; Mansur et al., 2019; McCormack et al., 2015; Murawski et al., 2014; O’Neill et al., 2019; Oloff et al., 2015; Sayana & Maffulli, 2007; Silbernagel et al., 2007; Stenson et al., 2018; Syvertson et al., 2017; Von Wehren et al., 2019; Wei et al., 2017; Welsh & Clodman, 1980; Zellers et al., 2019; Zhang et al., 2020; Zhuang et al., 2019) using it as a criteria to diagnose both midportion and insertional Achilles tendinopathy. Midportion tendinopathy was defined as an area 2–4 cm above the calcaneal insertion in two studies (Crill, Berlet & Hyer, 2014; Welsh & Clodman, 1980), 2–6 cm above the calcaneal insertion in six studies (Chester et al., 2008; Knobloch, 2007; Knobloch et al., 2008; Maffulli et al., 2008; Sayana & Maffulli, 2007; Syvertson et al., 2017), and 2–7 cm above the calcaneal insertion in one study (Murawski et al., 2014). Insertional tendinopathy was defined as the distal 2 cm in three studies (Cheng, Zhang & Cai, 2016; Mansur et al., 2017; McCormack et al., 2015), and the Achilles ‘insertion’ in five studies (Knobloch, 2007; Stenson et al., 2018; Zellers et al., 2019; Zhang et al., 2020; Zhuang et al., 2019). Pain with tendon loading was utilised as a diagnostic criteria in 18 studies (Alfredson & Spang, 2018; Barge-Caballero et al., 2008; Cheng, Zhang & Cai, 2016; Florit et al., 2019; Khan et al., 2003; Knobloch, 2007; Maffulli et al., 2008; McCormack et al., 2015; Murawski et al., 2014; O’Neill et al., 2019; Paavola et al., 2002; Paavola et al., 2000; Sayana & Maffulli, 2007; Silbernagel et al., 2007; Syvertson et al., 2017; Von Wehren et al., 2019; Wei et al., 2017; Welsh & Clodman, 1980), and duration of symptoms was utilised in 16 studies (Cheng, Zhang & Cai, 2016; Chester et al., 2008; Duthon et al., 2011; Jowett, Richmond & Bedi, 2018; Karjalainen et al., 2000; Knobloch et al., 2008; Lakshmanan & O’Doherty, 2004; McCormack et al., 2015; O’Neill et al., 2019; Oloff et al., 2015; Paavola et al., 2000; Silbernagel et al., 2007; Stenson et al., 2018; Von Wehren et al., 2019; Wei et al., 2017; Zhang et al., 2020). Duration of symptoms varied significantly with studies defining tendinopathy as symptoms lasting less than 6 months (Paavola et al., 2000), more than 6 weeks (McCormack et al., 2015; Von Wehren et al., 2019), more than 2 months (Silbernagel et al., 2007), more than three months (Chester et al., 2008; Knobloch et al., 2008; O’Neill et al., 2019; Stenson et al., 2018; Zhang et al., 2020), more than six months (Cheng, Zhang & Cai, 2016; Jowett, Richmond & Bedi, 2018; Lakshmanan & O’Doherty, 2004; Oloff et al., 2015; Wei et al., 2017), and more than 1 year (Duthon et al., 2011). As with the previous studies, the most common objective test for diagnosing Achilles tendinopathy was palpation, with 23 studies utilising it as a diagnostic criteria (Barge-Caballero et al., 2008; Chester et al., 2008; Crill, Berlet & Hyer, 2014; Duthon et al., 2011; Florit et al., 2019; Jowett, Richmond & Bedi, 2018; Karjalainen et al., 2000; Khan et al., 2003; Knobloch et al., 2008; Maffulli et al., 2008; Mansur et al., 2019; McCormack et al., 2015; Murawski et al., 2014; O’Neill et al., 2019; Paavola et al., 2002; Paavola et al., 2000; Sayana & Maffulli, 2007; Syvertson et al., 2017; Von Wehren et al., 2019; Wei et al., 2017; Welsh & Clodman, 1980; Zellers et al., 2019; Zhuang et al., 2019). Six studies used only subjective measures for diagnosing Achilles tendinopathy (Alfredson & Spang, 2018; Cheng, Zhang & Cai, 2016; Knobloch, 2007; Lakshmanan & O’Doherty, 2004; Oloff et al., 2015; Zhang et al., 2020).

Case-control studies

Of the 30 case-control studies, one (Chimenti et al., 2016) investigated insertional Achilles tendinopathy, 15 studies (Hutchison et al., 2013; Abate & Salini, 2019; Azevedo et al., 2009; Creaby et al., 2017; Gärdin et al., 2016; Lohrer & Nauck, 2009; Maffulli et al., 2003; Nadeau et al., 2016; Neeter et al., 2003; Pingel et al., 2013; Reid et al., 2012; Romero-Morales et al., 2019a; Ryan et al., 2009; Sengkerij et al., 2009; Romero-Morales et al., 2019b) investigated midportion Achilles tendinopathy, nine studies (Chimenti et al., 2020; Coombes et al., 2018; Eckenrode, Kietrys & Stackhouse, 2019; Hernández-Sánchez et al., 2018; Ooi et al., 2015; Rabello et al., 2020; Reiter et al., 2004; Verrall, Schofield & Brustad, 2011; Zhang et al., 2017) investigated both insertional and midportion Achilles tendinopathy, with five studies (Cassel et al., 2018; Holmes & Lin, 2006; Jewson et al., 2017; Leung & Griffith, 2008; De Mesquita et al., 2018) not specifying tendinopathy location (Table 8). As with the previous study types, the most commonly used diagnostic feature was location of pain, which was utilised in 27 of the case-control studies (Hutchison et al., 2013; Abate & Salini, 2019; Chimenti et al., 2016; Chimenti et al., 2020; Coombes et al., 2018; Creaby et al., 2017; Eckenrode, Kietrys & Stackhouse, 2019; Gärdin et al., 2016; Hernández-Sánchez et al., 2018; Holmes & Lin, 2006; Jewson et al., 2017; Leung & Griffith, 2008; Lohrer & Nauck, 2009; Nadeau et al., 2016; Neeter et al., 2003; De Mesquita et al., 2018; Ooi et al., 2015; Pingel et al., 2013; Rabello et al., 2020; Reid et al., 2012; Reiter et al., 2004; Romero-Morales et al., 2019a; Ryan et al., 2009; Sengkerij et al., 2009; Verrall, Schofield & Brustad, 2011; Zhang et al., 2017; Romero-Morales et al., 2019b). Insertional tendinopathy was defined as the distal 2 cm of the Achilles tendon in three studies (Chimenti et al., 2020; Rabello et al., 2020; Verrall, Schofield & Brustad, 2011). Midportion tendinopathy was defined as 2–6 cm above the calcaneal insertion in eight studies (Hutchison et al., 2013; Chimenti et al., 2020; Neeter et al., 2003; Rabello et al., 2020; Reid et al., 2012; Ryan et al., 2009; Verrall, Schofield & Brustad, 2011; Zhang et al., 2017), 2–7 cm above the calcaneal insertion in three studies (Gärdin et al., 2016; Lohrer & Nauck, 2009; Sengkerij et al., 2009), and the middle third of the tendon in one study (Nadeau et al., 2016). Additionally, duration of symptoms was commonly used to diagnose Achilles tendinopathy, with variations in the criteria. Achilles tendinopathy was defined as duration of symptoms of less than three months in one study (Neeter et al., 2003), greater than four weeks in two studies (Jewson et al., 2017; Nadeau et al., 2016), greater than two months in one study (De Mesquita et al., 2018), greater than three months in eight studies (Chimenti et al., 2020; Coombes et al., 2018; Eckenrode, Kietrys & Stackhouse, 2019; Ooi et al., 2015; Romero-Morales et al., 2019a; Ryan et al., 2009; Verrall, Schofield & Brustad, 2011; Romero-Morales et al., 2019b), and greater than six months in three studies (Gärdin et al., 2016; Leung & Griffith, 2008; Pingel et al., 2013). Pain with tendon loading was included as a diagnostic criteria in 18 studies (Abate & Salini, 2019; Azevedo et al., 2009; Cassel et al., 2018; Chimenti et al., 2016; Chimenti et al., 2020; Creaby et al., 2017; Eckenrode, Kietrys & Stackhouse, 2019; Hernández-Sánchez et al., 2018; Holmes & Lin, 2006; Jewson et al., 2017; Lohrer & Nauck, 2009; Reid et al., 2012; Reiter et al., 2004; Romero-Morales et al., 2019a; Ryan et al., 2009; Sengkerij et al., 2009; Verrall, Schofield & Brustad, 2011; Romero-Morales et al., 2019b). One study did not specify a subjective criteria to diagnose Achilles tendinopathy (Maffulli et al., 2003). Similar to previous study designs, palpation was the most common clinical test to diagnose Achilles tendinopathy, with it being used in 26 studies (Hutchison et al., 2013; Abate & Salini, 2019; Azevedo et al., 2009; Cassel et al., 2018; Chimenti et al., 2016; Chimenti et al., 2020; Coombes et al., 2018; Creaby et al., 2017; Eckenrode, Kietrys & Stackhouse, 2019; Gärdin et al., 2016; Holmes & Lin, 2006; Leung & Griffith, 2008; Lohrer & Nauck, 2009; Maffulli et al., 2003; Nadeau et al., 2016; Neeter et al., 2003; De Mesquita et al., 2018; Ooi et al., 2015; Pingel et al., 2013; Reid et al., 2012; Reiter et al., 2004; Romero-Morales et al., 2019a; Sengkerij et al., 2009; Verrall, Schofield & Brustad, 2011; Zhang et al., 2017; Romero-Morales et al., 2019b). Four studies relied only on subjective measures to diagnose Achilles tendinopathy (Hernández-Sánchez et al., 2018; Jewson et al., 2017; Rabello et al., 2020; Ryan et al., 2009).

Table 8 Case-control studies.

Author	Year	Sample size	Location	Subjective history	Clinical tests	
Maffulli et al. (2003)	2003	24 (M/F = 24/0)	Midportion	Not specified	Pain on palpation
Painful Arc Sign
Royal London Hospital Test	
Neeter et al. (2003)	2003	25 (M/F = 15/10)	Midportion	Location of pain (2–6 cm above calcaneal insertion)
Duration of symptoms (<3 months)	Pain on palpation
Range of motion
Single-leg heel raise	
Reiter et al. (2004)	2004	35 (M/F = 30/5)	Insertional Midportion	Location of pain
Pain with tendon loading
VISA-A	Pain on palpation
Localised tendon thickening on palpation	
Holmes & Lin (2006)	2006	82 (M/F = 44/38)	Not specified	Location of pain
Pain with tendon loading	Pain on palpation	
Leung & Griffith (2008)	2008	71 (M/F = 31/40)	Not specified	Location of pain
Duration of symptoms (>6 months)	Pain on palpation
Localised thickening on palpation
Localised swelling on palpation	
Azevedo et al. (2009)	2009	42 (M/F = 32/10)	Midportion	Gradual onset of pain
Tendon stiffness
Swelling
Pain with tendon loading	Pain on palpation
Localised tendon thickening on palpation
Painful Arc Sign	
Lohrer & Nauck (2009)	2009	119 (M/F = not specified)	Midportion	Location of pain (2–7 cm above calcaneal insertion)
Pain with tendon loading	Pain on palpation	
Ryan et al. (2009)	2009	48 (M/F = 48/0)	Midportion	Location of pain (2–6 cm above calcaneal insertion)
Duration of symptoms (>3 months)
Pain with tendon loading	Not specified	
Sengkerij et al. (2009)	2009	25 (M/F = 16/9)	Midportion	Location of pain (2–7 cm above calcaneal insertion)
Pain with tendon loading	Pain on palpation	
Verrall, Schofield & Brustad (2011)	2011	190 (M/F = 108/82)	Insertional Midportion	Location of pain (2–6 cm above calcaneal insertion)
Location of pain (distal 2 cm)
Duration of symptoms (>3 months)
Pain with tendon loading
Tendon stiffness	Pain on palpation
Localised swelling on palpation	
Reid et al. (2012)	2012	36 (M/F = not specified)	Midportion	Location of pain (2–6 cm above calcaneal insertion)
Pain with tendon loading	Pain on palpation	
Hutchison et al. (2013)	2013	21 (M/F = 9/12)	Midportion	Location of pain (2–6 cm above calcaneal insertion)
Tendon stiffness	Pain on palpation
Localised tendon thickening on palpation
Painful Arc Sign
Royal London Hospital Test Pain with dorsiflexion
Single-leg heel raise Hopping	
Pingel et al. (2013)	2013	18 (M/F = 10/8)	Midportion	Location of pain
Duration of symptoms (>6 months)	Pain on palpation
Localised swelling on palpation	
Ooi et al. (2015)	2015	240 (M/F = 180/60)	Insertional Midportion	Location of pain
Duration of symptoms (>3 months)	Pain on palpation
Localised swelling on palpation	
Chimenti et al. (2016)	2016	40 (M/F = 20/20)	Insertional	Location of pain (insertion)
Pain with tendon loading	Pain on palpation	
Gärdin et al. (2016)	2016	30 (M/F = 12/18)	Midportion	Location of pain (2–7 cm above calcaneal insertion)
Duration of symptoms (>6 months)	Pain on palpation	
Nadeau et al. (2016)	2016	43 (M/F = 30/13)	Midportion	Location of pain (middle third)
Duration of symptoms (>4 weeks)
VISA-A	Pain on palpation
Localised tendon thickening on palpation Pain with passive dorsiflexion Pain with resisted plantarflexion
Single-leg heel raise
Hopping	
Creaby et al. (2017)	2017	25 (M/F = 25/0)	Midportion	Location of pain
Pain with tendon loading
Tendon stiffness	Pain on palpation
Hopping	
Jewson et al. (2017)	2017	35 (M/F = 22/13)	Not specified	Location of pain
Duration of symptoms (>4 weeks)
Pain with tendon loading	Not specified	
Zhang et al. (2017)	2017	37 (M/F = 26/11)	Insertional Midportion	Location of pain (2–6 cm above calcaneal insertion)
Location of pain (insertion)	Pain on palpation
Pain with resisted plantarflexion	
Cassel et al. (2018)	2018	182 (M/F = 113/69)	Not specified	Pain with tendon loading	Pain on palpation	
Coombes et al. (2018)	2018	67 (M/F = 37/30)	Insertional Midportion	Location of pain
Duration of symptoms (>3 months)	Pain on palpation
Single-leg heel raise	
Hernández-Sánchez et al. (2018)	2018	210 (M/F = 148/62)	Insertional Midportion	Location of pain
Pain with tendon loading
Tendon stiffness	Not specified	
De Mesquita et al. (2018)	2018	67 (M/F = 41/26)	Not specified	Location of pain
Duration of symptoms (>2 months)	Pain on palpation	
Abate & Salini (2019)	2019	64 (M/F = 40/24)	Midportion	Location of pain
Pain with tendon loading	Pain on palpation	
Eckenrode, Kietrys & Stackhouse (2019)	2019	41 (M/F = 19/22)	Insertional Midportion	Location of pain
Duration of symptoms (>3 months)
Pain with tendon loading	Pain on palpation
Single-leg heel raise	
Romero-Morales et al. (2019a)	2019	141 (M/F = 116/25)	Midportion	Location of pain
Duration of symptoms (>3 months)
Pain with tendon loading	Pain on palpation	
Romero-Morales et al. (2019b)	2019	143 (M/F = not specified)	Midportion	Location of pain
Duration of symptoms (>3 months)
Pain with tendon loading	Pain on palpation	
Chimenti et al. (2020)	2020	46 (M/F = 30/16)	Insertional Midportion	Location of pain (2–6 cm above calcaneal insertion)
Location of pain (distal 2 cm)
Duration of symptoms (>3 months) Pain with tendon loading
Tendon stiffness	Pain on palpation	
Rabello et al. (2020)	2020	46 (M/F = 30/16)	Insertional Midportion	Location of pain (2–6 cm above calcaneal insertion)
Location of pain (distal 2 cm)	Not specified	
Note:

cm, centimetres; M, male; F, female; VISA-A, Victorian Institute of Sport Assessment-Achilles.

Cross-sectional studies

Table 9 provides an overview of the 17 included cross-sectional studies, with 10 studies (De Jonge et al., 2011; De Marchi et al., 2018; Divani et al., 2010; Finnamore et al., 2019; Maffulli et al., 2008; Praet et al., 2018; Santamato et al., 2019; Scholes et al., 2018; Van der Vlist et al., 2020; Wang et al., 2012) investigating midportion Achilles tendinopathy, four studies (Docking et al., 2015; Kragsnaes et al., 2014; Turner et al., 2020; Vallance et al., 2020) investigating both insertional and midportion Achilles tendinopathy, and three studies (Aiyegbusi, Tella & Sanusi, 2020; Longo et al., 2009; Mantovani et al., 2020) not specifying tendinopathy location. Once again, location of pain was the most common subjective measure to diagnose Achilles tendinopathy, with 12 studies utilising it as a diagnostic criteria (De Jonge et al., 2011; De Marchi et al., 2018; Kragsnaes et al., 2014; Maffulli et al., 2008; Mantovani et al., 2020; Praet et al., 2018; Chester et al., 2008; Scholes et al., 2018; Turner et al., 2020; Vallance et al., 2020; Van der Vlist et al., 2020; Wang et al., 2012). Midportion Achilles tendinopathy was defined as 2–6 cm above the calcaneal insertion in four studies (De Marchi et al., 2018; Maffulli et al., 2008; Praet et al., 2018; Wang et al., 2012), and 2–7 cm above the calcaneal insertion in one study (Van der Vlist et al., 2020). Similarly, 12 studies included duration of symptoms as a diagnostic criteria, with durations of symptoms including greater than four weeks (Santamato et al., 2019), greater than two months (Praet et al., 2018; Van der Vlist et al., 2020), greater than three months (Finnamore et al., 2019; Maffulli et al., 2008; Mantovani et al., 2020; Scholes et al., 2018; Turner et al., 2020; Vallance et al., 2020; Wang et al., 2012), greater than four months (Kragsnaes et al., 2014), and greater than six months (De Marchi et al., 2018). Pain with tendon loading was the next most common subjective diagnostic measure, with nine studies including it as a diagnostic measure (De Marchi et al., 2018; Docking et al., 2015; Finnamore et al., 2019; Maffulli et al., 2008; Scholes et al., 2018; Turner et al., 2020; Vallance et al., 2020; Van der Vlist et al., 2020; Wang et al., 2012). One study (Divani et al., 2010) did not report subjective measures to confirm the diagnosis of Achilles tendinopathy. The most common clinical test included palpation, with nine studies using palpation to clinically diagnose Achilles tendinopathy (Divani et al., 2010; Finnamore et al., 2019; Kragsnaes et al., 2014; Longo et al., 2009; Maffulli et al., 2008; Mantovani et al., 2020; Praet et al., 2018; Van der Vlist et al., 2020; Wang et al., 2012). Four studies (Aiyegbusi, Tella & Sanusi, 2020; Longo et al., 2009; Santamato et al., 2019; Wang et al., 2012) included the Royal London Hospital Test as a clinical measure of Achilles tendinopathy, with four studies (De Jonge et al., 2011; De Marchi et al., 2018; Scholes et al., 2018; Turner et al., 2020) not specifying the clinical tests utilised to confirm the diagnosis.

Table 9 Cross-sectional studies.

Author	Year	Sample size	Location	Subjective history	Clinical tests	
Maffulli et al. (2008)	2008	50 (M/F = 50/0)	Midportion	Location of pain (2–6 cm above calcaneal insertion)
Duration of symptoms (>3 months)
Pain with tendon loading	Pain on palpation
Localised swelling on palpation	
Longo et al. (2009)	2009	178 (M/F = 110/68)	Not specified	VISA-A	Pain on palpation
Localised tendon thickening on palpation
Painful Arc Sign
Royal London Hospital Test	
Divani et al. (2010)	2010	26 (M/F = 17/9)	Midportion	Not specified	Pain on palpation	
De Jonge et al. (2011)	2011	107 (M/F = 51/56)	Midportion	Location of pain (above calcaneal insertion)	Not specified	
Wang et al. (2012)	2012	17 (M/F = 17/0)	Midportion	Location of pain (2–6 cm above calcaneal insertion)
Duration of symptoms (>3 months)
Pain with tendon loading
VISA-A	Pain on palpation
Royal London Hospital Test	
Kragsnaes et al. (2014)	2014	50 (M/F = 27/23)	Insertional Midportion	Location of pain
Duration of symptoms (>4 months)	Pain on palpation	
Docking et al. (2015)	2015	21 (M/F = 20/1)	Insertional Midportion	Pain with tendon loading	Single-leg heel raise
Hopping	
De Marchi et al. (2018)	2018	27 (M/F = 19/8)	Midportion	Location of pain (2–6 cm above calcaneal insertion)
Duration of symptoms (>6 months)
Pain with tendon loading	Not specified	
Praet et al. (2018)	2018	20 (M/F = 13/7)	Midportion	Location of pain (2–6 cm above calcaneal insertion)
Duration of symptoms (>2 months)	Pain on palpation	
Scholes et al. (2018)	2018	21 (M/F = 21/0)	Midportion	Location of pain (midportion)
Duration of symptoms (>3 months)
Pain with tendon loading
Tendon stiffness	Not specified	
Finnamore et al. (2019)	2019	25 (M/F = 12/13)	Midportion	Duration of symptoms (>3 months)
Pain with tendon loading	Pain on palpation	
Santamato et al. (2019)	2019	12 (M/F = 7/5)	Midportion	Location of pain
Duration of symptoms (>4 weeks)	Reduced ROM Pain during AROM
Painful Arc Sign
Royal London Hospital Test	
Aiyegbusi, Tella & Sanusi (2020)	2020	85 (M/F = 56/29)	Not specified	VISA-A	Royal London Hospital Test	
Mantovani et al. (2020)	2020	19 (M/F = 13/6)	Not specified	Location of pain
Duration of symptoms (>3 months)
VISA-A (<80)	Pain on palpation	
Turner et al. (2020)	2020	15 (M/F = 8/7)	Insertional Midportion	Location of pain
Duration of symptoms (>3 months)
Gradual onset of pain
Pain with tendon loading	Not specified	
Vallance et al. (2020)	2020	86 (M/F = 86/0)	Insertional Midportion	Location of pain
Duration of symptoms (>3 months)
Pain with tendon loading
Tendon stiffness	Single-leg heel raise
Hopping	
Van der Vlist et al. (2020)	2020	28 (M/F = 16/12)	Midportion	Location of pain (2–7 cm above calcaneal insertion)
Duration of symptoms (>2 months) Pain with tendon loading	Pain on palpation
Localised swelling on palpation	
Note:

cm, centimetres; M, male; F, female; VISA-A, Victorian Institute of Sport Assessment-Achilles.

Narrative reviews

Of the 43 narrative reviews included in the scoping review, seven studies investigated insertional Achilles tendinopathy (Aldridge, 2004; Benazzo, Todesca & Ceciliani, 1997; Chimenti et al., 2017; Den Hartog, 2009; Hu & Flemister, 2008; Irwin, 2010; Maffulli et al., 2019), 18 studies investigated midportion Achilles tendinopathy (Alfredson, 2003; Alfredson & Cook, 2007; Bains & Porter, 2006; Courville, Coe & Hecht, 2009; Feilmeier, 2017; Järvinen et al., 2001; Kader et al., 2002; Longo, Ronga & Maffulli, 2009; Maffulli & Kader, 2002; Maffulli et al., 2012; Maffulli, Sharma & Luscombe, 2004; Maffulli, Via & Oliva, 2014; McShane, Ostick & McCabe, 2007; Paavola et al., 2002; Scott, Huisman & Khan, 2011; Sharma & Maffulli, 2006; Simpson & Howard, 2009; Tan & Chan, 2008), 13 studies investigated both insertional and midportion Achilles tendinopathy (Aronow, 2005; Asplund & Best, 2013; Bhatty, Khan & Zubairy, 2019; Chazan, 1998; Cook, Khan & Purdam, 2002; Fredericson, 1996; Furia & Rompe, 2007; Horn & McCollum, 2015; Jukes, Scott & Solan, 2020; Maffulli, Giai Via & Oliva, 2015; Maffulli et al., 2020; Pedowitz & Beck, 2017; Saini et al., 2015), and five studies did not specify tendinopathy location (Table 10) (Millar et al., 2021; Baskerville et al., 2018; Maffulli, Giuseppe Longo & Denaro, 2012; Nichols, 1989; Sorosky et al., 2004). The most common subjective diagnostic criteria for diagnosing Achilles tendinopathy was pain with tendon loading, with all 43 included reviews utilising as a diagnostic criteria (Millar et al., 2021; Aldridge, 2004; Alfredson, 2003; Alfredson & Cook, 2007; Aronow, 2005; Asplund & Best, 2013; Bains & Porter, 2006; Baskerville et al., 2018; Benazzo, Todesca & Ceciliani, 1997; Bhatty, Khan & Zubairy, 2019; Chazan, 1998; Chimenti et al., 2017; Cook, Khan & Purdam, 2002; Courville, Coe & Hecht, 2009; Den Hartog, 2009; Feilmeier, 2017; Fredericson, 1996; Furia & Rompe, 2007; Horn & McCollum, 2015; Hu & Flemister, 2008; Irwin, 2010; Järvinen et al., 2001; Jukes, Scott & Solan, 2020; Kader et al., 2002; Longo, Ronga & Maffulli, 2009; Maffulli, Giai Via & Oliva, 2015; Maffulli, Giuseppe Longo & Denaro, 2012; Maffulli & Kader, 2002; Maffulli et al., 2020; Maffulli et al., 2012; Maffulli et al., 2019; Maffulli, Sharma & Luscombe, 2004; Maffulli, Via & Oliva, 2014; McShane, Ostick & McCabe, 2007; Nichols, 1989; Paavola et al., 2002; Pedowitz & Beck, 2017; Saini et al., 2015; Scott, Huisman & Khan, 2011; Sharma & Maffulli, 2006; Simpson & Howard, 2009; Sorosky et al., 2004; Tan & Chan, 2008). Location of pain was included as a diagnostic criteria of Achilles tendinopathy in 31 studies, with midportion tendinopathy defined as ‘midportion’ in two studies (Saini et al., 2015; Tan & Chan, 2008), distal 5 cm of the Achilles tendon in one study (Nichols, 1989), 2–5 cm above the calcaneal insertion in three studies (Bains & Porter, 2006; Furia & Rompe, 2007; Simpson & Howard, 2009), 2–6 cm above the calcaneal insertion in 13 studies (Alfredson, 2003; Asplund & Best, 2013; Feilmeier, 2017; Fredericson, 1996; Horn & McCollum, 2015; Kader et al., 2002; Longo, Ronga & Maffulli, 2009; Maffulli, Giai Via & Oliva, 2015; Maffulli & Kader, 2002; Maffulli, Sharma & Luscombe, 2004; Maffulli, Via & Oliva, 2014; Pedowitz & Beck, 2017; Sharma & Maffulli, 2006), and 4–6 cm above the calcaneal insertion in one study (Maffulli et al., 2012). The third most common subjective criteria reported was tendon stiffness, with 24 studies including it as a diagnostic criteria (Millar et al., 2021; Alfredson, 2003; Alfredson & Cook, 2007; Asplund & Best, 2013; Bains & Porter, 2006; Baskerville et al., 2018; Benazzo, Todesca & Ceciliani, 1997; Chazan, 1998; Chimenti et al., 2017; Cook, Khan & Purdam, 2002; Courville, Coe & Hecht, 2009; Feilmeier, 2017; Fredericson, 1996; Hu & Flemister, 2008; Irwin, 2010; Jukes, Scott & Solan, 2020; Kader et al., 2002; Maffulli, Giuseppe Longo & Denaro, 2012; Maffulli et al., 2020; Maffulli et al., 2019; McShane, Ostick & McCabe, 2007; Nichols, 1989; Pedowitz & Beck, 2017; Tan & Chan, 2008). As with previous study types, the most common clinical test used to diagnose Achilles tendinopathy was palpation, with all 43 included reviews including it as a clinical measure (Millar et al., 2021; Aldridge, 2004; Alfredson, 2003; Alfredson & Cook, 2007; Aronow, 2005; Asplund & Best, 2013; Bains & Porter, 2006; Baskerville et al., 2018; Benazzo, Todesca & Ceciliani, 1997; Bhatty, Khan & Zubairy, 2019; Chazan, 1998; Chimenti et al., 2017; Cook, Khan & Purdam, 2002; Courville, Coe & Hecht, 2009; Den Hartog, 2009; Feilmeier, 2017; Fredericson, 1996; Furia & Rompe, 2007; Horn & McCollum, 2015; Hu & Flemister, 2008; Irwin, 2010; Järvinen et al., 2001; Jukes, Scott & Solan, 2020; Kader et al., 2002; Longo, Ronga & Maffulli, 2009; Maffulli, Giai Via & Oliva, 2015; Maffulli, Giuseppe Longo & Denaro, 2012; Maffulli & Kader, 2002; Maffulli et al., 2020; Maffulli et al., 2012; Maffulli et al., 2019; Maffulli, Sharma & Luscombe, 2004; Maffulli, Via & Oliva, 2014; McShane, Ostick & McCabe, 2007; Nichols, 1989; Paavola et al., 2002; Pedowitz & Beck, 2017; Saini et al., 2015; Scott, Huisman & Khan, 2011; Sharma & Maffulli, 2006; Simpson & Howard, 2009; Sorosky et al., 2004; Tan & Chan, 2008). There was then significant variation in other clinical tests used to diagnose Achilles tendinopathy, with nine studies including the Painful Arc Sign (Millar et al., 2021; Aronow, 2005; Feilmeier, 2017; Horn & McCollum, 2015; Kader et al., 2002; Longo, Ronga & Maffulli, 2009; Maffulli, Giuseppe Longo & Denaro, 2012; Maffulli & Kader, 2002; Maffulli et al., 2020), seven studies including reduced range of motion (Bhatty, Khan & Zubairy, 2019; Chazan, 1998; Chimenti et al., 2017; Furia & Rompe, 2007; Hu & Flemister, 2008; Maffulli et al., 2012; Nichols, 1989), six studies including the Royal London Hospital Test (Millar et al., 2021; Feilmeier, 2017; Horn & McCollum, 2015; Jukes, Scott & Solan, 2020; Longo, Ronga & Maffulli, 2009; Maffulli et al., 2020), and six studies including pain whilst hopping as a clinical diagnostic criteria (Millar et al., 2021; Alfredson & Cook, 2007; Bains & Porter, 2006; Cook, Khan & Purdam, 2002; Feilmeier, 2017; Nichols, 1989).

Table 10 Narrative reviews.

Author	Year	Location	Subjective history	Clinical tests	
Nichols (1989)	1989	Not specified	Location of pain (Distal 5 cm)
Tendon stiffness
Pain with tendon loading
Gradual onset of pain
Change in activity	Pain on palpation
Localised tendon thickening on palpation
Localised swelling on palpation
Reduced ROM
Hopping	
Fredericson (1996)	1996	Insertional Midportion	Location of pain (2–6 cm above calcaneal insertion)
Location of pain (insertion)
Pain with tendon loading
Tendon stiffness	Pain on palpation
Localised tendon thickening on palpation
Localised swelling on palpation	
Benazzo, Todesca & Ceciliani (1997)	1997	Insertional	Pain with tendon loading
Tendon stiffness	Pain on palpation	
Chazan (1998)	1998	Insertional Midportion	Gradual onset of pain
Tendon stiffness
Pain with tendon loading	Pain on palpation
Localised tendon thickening on palpation
Localised swelling on palpation
Pain with passive dorsiflexion
Pain with resisted plantarflexion
Reduced ROM	
Järvinen et al. (2001)	2001	Midportion	Location of pain
Pain with tendon loading	Pain on palpation
Localised swelling on palpation	
Cook, Khan & Purdam (2002)	2002	Insertional Midportion	Gradual onset of pain
Location of pain
Pain with tendon loading
Tendon stiffness
Change in activity
VISA-A	Pain on palpation
Localised tendon thickening on palpation
Localised swelling on palpation
Single-leg heel raise
Hopping	
Kader et al. (2002)	2002	Midportion	Gradual onset of pain
Location of pain (2–6 cm above calcaneal insertion)
Duration of symptoms
Tendon stiffness
Pain with tendon loading	Pain on palpation
Localised tendon thickening on palpation
Painful Arc Sign	
Maffulli & Kader (2002)	2002	Midportion	Gradual onset of pain
Location of pain (2–6 cm above calcaneal insertion)
Duration of symptoms
Pain with tendon loading	Pain on palpation
Localised tendon thickening on palpation
Localised swelling on palpation
Painful Arc Sign	
Paavola et al. (2002)	2002	Midportion	Pain with tendon loading
Duration of symptoms	Pain on palpation
Localised tendon thickening on palpation
Localised swelling on palpation	
Alfredson (2003)	2003	Midportion	Location of pain (2–6 cm above calcaneal insertion)
Pain with tendon loading Tendon stiffness	Pain on palpation
Localised swelling on palpation	
Aldridge (2004)	2004	Insertional	Pain with tendon loading	Pain on palpation
Pain with passive dorsiflexion	
Maffulli, Sharma & Luscombe (2004)	2004	Midportion	Location of pain (2–6 cm above calcaneal insertion)
Pain with tendon loading	Pain on palpation
Localised tendon thickening on palpation
Localised swelling on palpation	
Sorosky et al. (2004)	2004	Not specified	Gradual onset of pain
Pain with tendon loading
Change in training	Pain on palpation
Localised tendon thickening on palpation
Localised swelling on palpation
Pain with passive dorsiflexion Pain with resisted plantarflexion	
Aronow (2005)	2005	Insertional Midportion	Pain with tendon loading	Pain on palpation
Localised swelling on palpation
Pain with passive dorsiflexion (insertional)
Painful Arc Sign (midportion)
Single-leg Heel Raise	
Bains & Porter (2006)	2006	Midportion	Gradual onset of pain
Location of pain (2–5 cm above calcaneal insertion)
Tendon stiffness
Pain with tendon loading
Change in training
VISA-A	Pain on palpation
Localised tendon thickening on palpation
Localised swelling on palpation
Hopping on the spot
Forward hopping 6m hop test	
Sharma & Maffulli (2006)	2006	Midportion	Location of pain (2–6 cm above calcaneal insertion)
Pain with tendon loading
Swelling	Pain on palpation
Localised tendon thickening on palpation
Localised swelling on palpation	
Alfredson & Cook (2007)	2007	Midportion	Pain with tendon loading
Tendon stiffness	Localised swelling on palpation
Single-leg Heel Raise
Hopping on the spot
Forward hopping	
Furia & Rompe (2007)	2007	Insertional Midportion	Location of pain (2–5 cm above calcaneal insertion) - midportion
Location of pain (insertion)
Pain with tendon loading	Pain on palpation
Localised swelling on palpation
Pain with passive dorsiflexion (insertional)
Reduced ROM	
McShane, Ostick & McCabe (2007)	2007	Midportion	Location of pain
Pain with tendon loading
Tendon stiffness	Pain on palpation
Localised tendon thickening on palpation Single-leg heel raise	
Hu & Flemister (2008)	2008	Insertional	Location of pain
Pain with tendon loading
Tendon stiffness	Pain on palpation
Localised tendon thickening on palpation Pain with passive dorsiflexion
Reduced ROM
Silfverskiold test	
Tan & Chan (2008)	2008	Midportion	Location of pain (midportion)
Duration of pain (>2 weeks)
Pain with tendon loading
Tendon stiffness	Pain on palpation
Localised tendon thickening on palpation
Localised swelling on palpation
Single-leg heel raise	
Courville, Coe & Hecht (2009)	2009	Midportion	Pain with tendon loading
Location of pain
Tendon stiffness
Change in activity	Pain on palpation
Localised tendon thickening on palpation
Localised swelling on palpation	
Den Hartog (2009)	2009	Insertional	Gradual onset of pain
Duration of symptoms
Pain with tendon loading	Pain on palpation
Localised tendon thickening on palpation
Localised swelling on palpation	
Longo, Ronga & Maffulli (2009)	2009	Midportion	Location of pain (2–6 cm above calcaneal insertion)
Pain with tendon loading
Swelling	Pain on palpation
Localised tendon thickening on palpation
Painful Arc Sign
Royal London Hospital Test	
Simpson & Howard (2009)	2009	Midportion	Location of pain (2–5 cm above calcaneal insertion)
Pain with tendon loading
Swelling
Change in activity	Pain on palpation
Localised swelling on palpation
Reduced flexibility in hamstring and calf	
Irwin (2010)	2010	Insertional	Location of pain (calcaneal tuberosity)
Swelling
Pain with tendon loading
Tendon stiffness	Pain on palpation
Localised swelling on palpation	
Scott, Huisman & Khan (2011)	2011	MIdportion	Location of pain
Pain with tendon loading
Swelling	Pain on palpation
Localised tendon thickening on palpation	
Maffulli et al. (2012)	2012	Midportion	Location of pain (4–6 cm above calcaneal insertion)
Pain with tendon loading
Swelling
Change in activity	Pain on palpation
Localised tendon thickening on palpation
Localised swelling on palpation
Reduced ROM	
Maffulli, Giuseppe Longo & Denaro (2012)	2012	Not specified	Gradual onset of pain
Tendon stiffness
Pain with tendon loading
Swelling
VISA-A	Pain on palpation
Localised tendon thickening on palpation
Localised swelling on palpation
Painful Arc Sign	
Asplund & Best (2013)	2013	Insertional Midportion	Location of pain (2–6 cm above calcaneal insertion)
Location of pain (insertion)
Pain with tendon loading
Tendon stiffness	Pain on palpation
Localised swelling on palpation	
Maffulli, Via & Oliva (2014)	2014	Midportion	Location of pain (2–6 cm above calcaneal insertion)
Pain with tendon loading	Pain on palpation
Localised swelling on palpation
Single-leg heel raise	
Horn & McCollum (2015)	2015	Insertional Midportion	Location of pain (2–6 cm above calcaneal insertion)
Location of pain (insertion)
Pain with tendon loading	Localised swelling on palpation
Painful Arc Sign
Royal London Hospital Test	
Maffulli, Giai Via & Oliva (2015)	2015	Insertional Midportion	Location of pain (2–6 cm above calcaneal insertion)
Location of pain (insertion)
Swelling
Pain with tendon loading	Pain on palpation	
Saini et al. (2015)	2015	Insertional Midportion	Location of pain (midportion)
Location of pain (insertion)
Pain with tendon loading	Pain on palpation
Localised swelling on palpation
Pain with dorsiflexion and plantarflexion	
Chimenti et al. (2017)	2017	Insertional	Location of pain (distal 2 cm)
Pain with tendon loading
Tendon stiffness	Pain on palpation
Localised swelling on palpation
Pain with passive dorsiflexion
Pain with resisted plantarflexion
Reduced ROM	
Feilmeier (2017)	2017	Midportion	Location of pain (2–6 cm above calcaneal insertion)
Swelling
Pain with tendon loading
Tendon stiffness
Change in activity
VISA-A	Pain on palpation
Localised tendon thickening on palpation
Pain with passive dorsiflexion
Painful Arc Sign
Royal London Hospital Test
Single-leg Heel Raise
Hopping	
Pedowitz & Beck (2017)	2017	Insertional Midportion	Location of pain (2–6 cm above calcaneal insertion)
Location of pain (insertion)
Pain with tendon loading
Tendon stiffness	Pain on palpation
Single leg heel raise	
Baskerville et al. (2018)	2018	Not specified	Gradual onset of pain
Pain with tendon loading
Tendon stiffness	Pain on palpation
Pain with passive and active movement Reduced strength	
Bhatty, Khan & Zubairy (2019)	2019	Insertional Midportion	Pain with tendon loading	Pain on palpation
Localised swelling on palpation
Pain with passive dorsiflexion (insertional)
Pain with resisted plantarflexion
Reduced ROM	
Maffulli et al. (2019)	2019	Insertional	Location of pain (distal 2 cm)
Pain with tendon loading
Tendon stiffness	Pain on palpation
Localised tendon thickening on palpation
Localised swelling on palpation	
Jukes, Scott & Solan (2020)	2020	Insertional Midportion	Location of pain
Pain with tendon loading
Tendon stiffness Duration of symptoms	Pain on palpation
Localised tendon thickening on palpation
Localised swelling on palpation
Royal London Hospital Test
Silfverskiold test	
Maffulli et al. (2020)	2020	Insertional Midportion	Location of pain
Pain with tendon loading
Localised swelling
Tendon stiffness	Pain on palpation
Painful Arc Sign
Royal London Hospital Test	
Millar et al. (2021)	2021	Not specified	Location of pain
Pain with tendon loading
Tendon stiffness	Pain on palpation
Painful Arc Sign
Royal London Hospital Test
Single leg heel raise
Hopping	
Note:

cm, centimetres; M, male; F, female; VISA-A, Victorian Institute of Sport Assessment-Achilles; ROM, range of motion; m, metres.

Case-reports

Table 11 highlights the characteristics of the included case report studies. Five studies (Borda & Selhorst, 2017; Jayaseelan, Weber & Jonely, 2019; Maffulli et al., 2011; Papa, 2012; Van Sterkenburg et al., 2011) investigated midportion Achilles tendinopathy and one study (Benito, 2016) investigated insertional Achilles tendinopathy. As with the narrative reviews, the most common subjective measure used to diagnose Achilles tendinopathy was pain with tendon loading (Benito, 2016; Borda & Selhorst, 2017; Jayaseelan, Weber & Jonely, 2019; Maffulli et al., 2011; Papa, 2012), with the second most common diagnostic criteria being location of pain (Benito, 2016; Jayaseelan, Weber & Jonely, 2019; Maffulli et al., 2011; Van Sterkenburg et al., 2011). Midportion Achilles tendinopathy was defined as a location of pain 2–4 cm above the calcaneal insertion in one study (Maffulli et al., 2011), and 4–7 cm above the calcaneal insertion in another study (Van Sterkenburg et al., 2011). Five studies (Borda & Selhorst, 2017; Jayaseelan, Weber & Jonely, 2019; Maffulli et al., 2011; Papa, 2012; Van Sterkenburg et al., 2011) used palpation as an objective measure for diagnosing Achilles tendinopathy, with three studies (Borda & Selhorst, 2017; Jayaseelan, Weber & Jonely, 2019; Papa, 2012) utilising the pain during single-leg heel raise and one study (Benito, 2016) not specifying any objective clinical tests.

Table 11 Case reports.

Author	Year	Sample size	Location	Subjective history	Clinical tests	
Maffulli et al. (2011)	2011	1 (M/F = 1/0)	Midportion	Location of pain (2–4 cm above calcaneal insertion)
Duration of symptoms (>3 months)
Pain with tendon loading	Pain on palpation
Painful Arc Sign
Royal London Hospital Test	
Van Sterkenburg et al. (2011)	2011	3 (M/F = 1/2)	Midportion	Location of pain (4–7 cm above calcaneal insertion)
Tendon stiffness	Pain on palpation	
Papa (2012)	2012	1 (M/F = 0/1)	Midportion	Gradual onset of pain
Pain with tendon loading	Pain on palpation
Localised tendon thickening on palpation
Localised swelling on palpation
Reduced ROM
Single-leg heel raise	
Benito (2016)	2016	5 (M/F = 2/3)	Insertional	Location of pain (insertion)
Gradual onset of pain
Pain with tendon loading	Not specified	
Borda & Selhorst (2017)	2017	1 (M/F = 0/1)	Midportion	Gradual onset of pain
Pain with tendon loading	Pain on palpation
MMT
Single-leg heel raise	
Jayaseelan, Weber & Jonely (2019)	2019	2 (M/F = 1/1)	Midportion	Location of pain
Pain with tendon loading	Pain on palpation
Single-leg heel raise
Hopping
Reduced ROM	
Note:

cm, centimetres; M, male; F, female; ROM, range of motion; MMT, manual muscle test.

Protocols

Of the five included protocol studies, two studies (Habets et al., 2017; Hasani et al., 2020) investigated midportion Achilles tendinopathy, one study (Mansur et al., 2017) investigated insertional Achilles tendinopathy and two studies (Barker-Davies et al., 2017; Post et al., 2020) investigated both insertional and midportion Achilles tendinopathy (Table 12). The most common reported subjective criteria utlised to diagnose Achilles tendinopathy was location of pain, with midportion Achilles tendinopathy defined as pain 2–6 cm above the calcaneal insertion in two studies (Barker-Davies et al., 2017; Hasani et al., 2020), and 2–7 cm above the calcaneal insertion in one study (Habets et al., 2017). Insertional tendinopathy was defined as the distal 2 cm of the Achilles tendon in one study (Mansur et al., 2017) and the Achilles ‘insertion’ in another study (Barker-Davies et al., 2017). Clinical diagnostic tests varied with three studies (Barker-Davies et al., 2017; Habets et al., 2017; Mansur et al., 2017) including palpation, two studies (Barker-Davies et al., 2017; Post et al., 2020) including pain during a single-leg heel raise and two studies including pain during hopping (Aggarwal et al., 2015; O’Neill et al., 2019). One study did not specify objective clinical tests (Hasani et al., 2020).

Table 12 Protocols.

Author	Year	Location	Subjective history	Clinical tests	
Barker-Davies et al. (2017)	2017	Insertional Midportion	Location of pain (2–6 cm above calcaneal insertion)
Location of pain (insertion)
Pain with tendon loading
Tendon stiffness
Change in training	Pain on palpation
Single-leg heel raise
Hopping	
Habets et al. (2017)	2017	Midportion	Location of pain (2–7 cm above calcaneal insertion)
Duration of symptoms (>3 months)
Pain with tendon loading	Pain on palpation
Localised swelling on palpation	
Mansur et al. (2017)	2017	Insertional	Location of pain (distal 2 cm)
Duration of symptoms (>3 months)	Pain on palpation	
Hasani et al. (2020)	2020	Midportion	Location of pain (2–6 cm above calcaneal insertion)
Pain with tendon loading
Tendon stiffness
VISA-A	Not specified	
Post et al. (2020)	2020	Insertional Midportion	Location of pain
Pain with tendon loading	Walking Single-leg heel raise Hopping	
Note:

cm, centimetres; VISA-A, Victorian Institute of Sport Assessment-Achilles.

Outcome measures

Within the 159 included articles there were 42 different outcome measures in the clinical diagnosis of Achilles tendinopathy, with 49 studies (Millar et al., 2021; Reiman et al., 2014; Aldridge, 2004; Alfredson & Cook, 2007; Aronow, 2005; Barge-Caballero et al., 2008; Baskerville et al., 2018; Benazzo, Todesca & Ceciliani, 1997; Bhatty, Khan & Zubairy, 2019; Bjordal, Lopes-Martins & Iversen, 2006; Cassel et al., 2018; Chazan, 1998; Chimenti et al., 2017; Courville, Coe & Hecht, 2009; De Jonge et al., 2011; Den Hartog, 2009; Docking et al., 2015; Florit et al., 2019; Fredericson, 1996; Furia & Rompe, 2007; Holmes & Lin, 2006; Horn & McCollum, 2015; Irwin, 2010; Järvinen et al., 2001; Jukes, Scott & Solan, 2020; Kader et al., 2002; Knobloch et al., 2008; Maffulli, Giai Via & Oliva, 2015; Maffulli & Kader, 2002; Maffulli et al., 2003; Maffulli et al., 2011; Maffulli et al., 2020; Maffulli et al., 2012; Maffulli, Sharma & Luscombe, 2004; Maffulli, Via & Oliva, 2014; McShane, Ostick & McCabe, 2007; Nichols, 1989; Paavola et al., 2002; Paavola et al., 2002; Pedowitz & Beck, 2017; Ryan et al., 2009; Saini et al., 2015; Sharma & Maffulli, 2006; Simpson & Howard, 2009; Sorosky et al., 2004; Thomas et al., 2010; Welsh & Clodman, 1980; Xu et al., 2019; Zhang et al., 2017) not reporting any outcome measures. Of the 110 included studies to report on outcome measures (Hutchison et al., 2013; Abate & Salini, 2019; Aiyegbusi, Tella & Sanusi, 2020; Alfredson, 2003; Alfredson & Spang, 2018; Asplund & Best, 2013; Azevedo et al., 2009; Bains & Porter, 2006; Barker-Davies et al., 2017; Benito, 2016; Boesen et al., 2017; Borda & Selhorst, 2017; Brown et al., 2006; Carcia et al., 2010; Cheng, Zhang & Cai, 2016; Chester et al., 2008; Chimenti et al., 2016; Chimenti et al., 2020; Cook, Khan & Purdam, 2002; Coombes et al., 2018; Creaby et al., 2017; Crill, Berlet & Hyer, 2014; De Marchi et al., 2018; Divani et al., 2010; Duthon et al., 2011; Ebbesen et al., 2018; Eckenrode, Kietrys & Stackhouse, 2019; Feilmeier, 2017; Finnamore et al., 2019; Gärdin et al., 2016; Gatz et al., 2020; Habets et al., 2017; Hasani et al., 2020; Hernández-Sánchez et al., 2018; Hu & Flemister, 2008; Hutchison et al., 2011; Jayaseelan, Weber & Jonely, 2019; Jewson et al., 2017; Jowett, Richmond & Bedi, 2018; Karjalainen et al., 2000; Khan et al., 2003; Knobloch, 2007; Knobloch et al., 2007; Kragsnaes et al., 2014; Krogh et al., 2016; Lakshmanan & O’Doherty, 2004; Leung & Griffith, 2008; Lohrer & Nauck, 2009; Longo et al., 2009; Longo, Ronga & Maffulli, 2009; Maffulli, Giuseppe Longo & Denaro, 2012; Maffulli et al., 2008; Maffulli et al., 2019; Maffulli et al., 2008; Mafi, Lorentzon & Alfredson, 2001; Magnussen, Dunn & Thomson, 2009; Mansur et al., 2019; Mansur et al., 2017; Mantovani et al., 2020; Martin et al., 2018; Mayer et al., 2007; McCormack et al., 2015; Murawski et al., 2014; Nadeau et al., 2016; Neeter et al., 2003; De Mesquita et al., 2018; O’Neill et al., 2019; Oloff et al., 2015; Ooi et al., 2015; Paavola et al., 2000; Paoloni et al., 2004; Papa, 2012; Petersen, Welp & Rosenbaum, 2007; Pingel et al., 2013; Post et al., 2020; Praet et al., 2018; Rabello et al., 2020; Rasmussen et al., 2008; Reid et al., 2012; Reiter et al., 2004; Romero-Morales et al., 2019a; Rompe, Furia & Maffulli, 2009; Rompe et al., 2008; Rompe et al., 2007; Roos et al., 2004; Santamato et al., 2019; Sayana & Maffulli, 2007; Scholes et al., 2018; Scott, Huisman & Khan, 2011; Sengkerij et al., 2009; Silbernagel et al., 2007; Silbernagel et al., 2001; Solomons et al., 2020; Stenson et al., 2018; Stergioulas et al., 2008; Syvertson et al., 2017; Tan & Chan, 2008; Turner et al., 2020; Vallance et al., 2020; Van der Vlist et al., 2020; Van der Vlist et al., 2020; Van Sterkenburg et al., 2011; Verrall, Schofield & Brustad, 2011; Von Wehren et al., 2019; Wang et al., 2012; Wei et al., 2017; Zellers et al., 2019; Zhang et al., 2020; Zhuang et al., 2019; Romero-Morales et al., 2019b), 42 different outcome measures were utilised. Disability was the most commonly measured outcome, with 28 different outcome measures for disability being applied 135 times (Fig. 4). The most common outcome measure for disability was the VISA-A questionnaire, being used in 75% of the studies reporting outcome meaures (Abate & Salini, 2019; Aiyegbusi, Tella & Sanusi, 2020; Alfredson & Spang, 2018; Asplund & Best, 2013; Bains & Porter, 2006; Barker-Davies et al., 2017; Benito, 2016; Boesen et al., 2017; Brown et al., 2006; Carcia et al., 2010; Cheng, Zhang & Cai, 2016; Chimenti et al., 2016; Chimenti et al., 2020; Cook, Khan & Purdam, 2002; Coombes et al., 2018; Creaby et al., 2017; Divani et al., 2010; Ebbesen et al., 2018; Eckenrode, Kietrys & Stackhouse, 2019; Feilmeier, 2017; Finnamore et al., 2019; Gärdin et al., 2016; Gatz et al., 2020; Habets et al., 2017; Hasani et al., 2020; Hernández-Sánchez et al., 2018; Hutchison et al., 2011; Jayaseelan, Weber & Jonely, 2019; Jewson et al., 2017; Jowett, Richmond & Bedi, 2018; Khan et al., 2003; Kragsnaes et al., 2014; Krogh et al., 2016; Lakshmanan & O’Doherty, 2004; Leung & Griffith, 2008; Lohrer & Nauck, 2009; Longo et al., 2009; Longo, Ronga & Maffulli, 2009; Maffulli, Giuseppe Longo & Denaro, 2012; Maffulli et al., 2008; Maffulli et al., 2019; Maffulli et al., 2008; Magnussen, Dunn & Thomson, 2009; Mansur et al., 2019; Mansur et al., 2017; Mantovani et al., 2020; Martin et al., 2018; McCormack et al., 2015; Nadeau et al., 2016; De Mesquita et al., 2018; O’Neill et al., 2019; Oloff et al., 2015; Ooi et al., 2015; Pingel et al., 2013; Praet et al., 2018; Rabello et al., 2020; Reid et al., 2012; Reiter et al., 2004; Romero-Morales et al., 2019a; Rompe, Furia & Maffulli, 2009; Rompe et al., 2008; Rompe et al., 2007; Santamato et al., 2019; Sayana & Maffulli, 2007; Scholes et al., 2018; Scott, Huisman & Khan, 2011; Sengkerij et al., 2009; Silbernagel et al., 2007; Solomons et al., 2020; Syvertson et al., 2017; Tan & Chan, 2008; Turner et al., 2020; Vallance et al., 2020; Van der Vlist et al., 2020; Van der Vlist et al., 2020; Van Sterkenburg et al., 2011; Von Wehren et al., 2019; Wang et al., 2012; Wei et al., 2017; Zellers et al., 2019; Zhang et al., 2020; Romero-Morales et al., 2019b). Following, disability, the second most common outcome measure was pain with the VAS (Hutchison et al., 2013; Alfredson, 2003; Alfredson & Spang, 2018; Barker-Davies et al., 2017; Boesen et al., 2017; Chester et al., 2008; Ebbesen et al., 2018; Habets et al., 2017; Hutchison et al., 2011; Knobloch, 2007; Knobloch et al., 2007; Maffulli et al., 2019; Mafi, Lorentzon & Alfredson, 2001; Magnussen, Dunn & Thomson, 2009; Mansur et al., 2019; Mansur et al., 2017; Petersen, Welp & Rosenbaum, 2007; Rasmussen et al., 2008; Reid et al., 2012; Romero-Morales et al., 2019a; Santamato et al., 2019; Scott, Huisman & Khan, 2011; Silbernagel et al., 2001; Stenson et al., 2018; Stergioulas et al., 2008; Turner et al., 2020; Van der Vlist et al., 2020; Van Sterkenburg et al., 2011; Verrall, Schofield & Brustad, 2011; Wei et al., 2017; Zhang et al., 2020; Zhuang et al., 2019; Romero-Morales et al., 2019b) being used in 30% of studies reporting outcome measures and numerical pain rating scale (NPRS) (Borda & Selhorst, 2017; Brown et al., 2006; Chimenti et al., 2016; Chimenti et al., 2020; Coombes et al., 2018; Hasani et al., 2020; Jayaseelan, Weber & Jonely, 2019; Krogh et al., 2016; Mantovani et al., 2020; O’Neill et al., 2019; Paoloni et al., 2004; Papa, 2012; Post et al., 2020; Rompe, Furia & Maffulli, 2009; Rompe et al., 2008; Rompe et al., 2007; Syvertson et al., 2017; Vallance et al., 2020) being used in 17% of studies reporting outcome measures. Outcome measures related to quality of life were utilised within 12 studies, with the most common outcome measures used being the 12-Item Short Form Survey (SF-12) (Duthon et al., 2011; Habets et al., 2017; Mansur et al., 2017; Murawski et al., 2014; Stenson et al., 2018), EuroQol 5 Dimension 5 Level Questionnaire (EQ-5D-5L) (Chester et al., 2008; De Marchi et al., 2018; Habets et al., 2017; Hasani et al., 2020) and 36-Item Short Form Survey (SF-36) (Hutchison et al., 2011; Maffulli et al., 2019; Petersen, Welp & Rosenbaum, 2007). Similarly, psychosocial outcomes were poorly measured, being utilised on 11 occasions, with the most common outcome measures being the Pain Catastrophizing Scale (PCS) (Chimenti et al., 2020; Eckenrode, Kietrys & Stackhouse, 2019; Hasani et al., 2020; Post et al., 2020; Vallance et al., 2020), Tampa Kinesiophobia Scale (TKS) (Chimenti et al., 2020; Hasani et al., 2020; Post et al., 2020; Vallance et al., 2020), Pain Disability Index (PDI) (Mayer et al., 2007; Scott, Huisman & Khan, 2011) and Pain Efficacy Scale (PES) (Mayer et al., 2007; Scott, Huisman & Khan, 2011).

Figure 4 Outcome measures used to measure Achilles tendinopathy categorised by purpose.

Discussion

Overview

The clinical diagnosis of tendinopathy is commonly determined via both patient history and clinical tests (Cook et al., 2016; Coombes, Bisset & Vicenzino, 2015; Lewis, 2016; Lewis et al., 2015; Maffulli, Khan & Puddu, 1998; Malliaras et al., 2015; Scott et al., 2013). However, with no consensus on gold standard clinical tests with which to diagnose tendinopathy (Docking, Ooi & Connell, 2015), many research studies utilise a variety of measures to diagnose Achilles tendinopathy (Hutchison et al., 2013). The primary aim of this scoping review was to provide a method for clinically diagnosing Achilles tendinopathy that aligns with the nine core health domains. In order to achieve this, specific objectives were determined that included identifying the most common clinical tests used to diagnose Achilles tendinopathy, identifying the most common outcome measures used to assess Achilles tendinopathy, and summarising the studies to date. This will allow for greater consistency in both research and clinical settings. Additionally, this review aimed to identify the both the areas of strength and weakness

Terminology

As highlighted in Fig. 3, ‘Tendinopathy’ was the most commonly term used to describe persistent Achilles tendon pain, particularly in more recent studies. This scoping review aligns with the previous consensus statements advocating the consistent use of the term tendinopathy to describe persistent Achilles tendon pain and associated loss of function in relation to mechanical loading (Scott et al., 2020). There was a noticeable reduction of the use of alternative terms such as tendinitis and tendinosis, particularly since 2018, indicating progression towards unifying the terminology used to describe the clinical condition of persistent pain and dysfunction in the Achilles tendon that is associated with mechanical loading.

A difficulty identified in this scoping review was the inclusion of symptom duration as a measure to diagnose Achilles tendinopathy (Tables 6, 7, 8 and 9). When used as a measure, duration of symptoms varied significantly from four weeks up to 12 months, making identifying a consistent duration of symptoms to diagnose Achilles tendinopathy difficult and potentially contributing to the different terminology used within research and clinical practice. The term tendinitis indicates an inflammatory condition of the Achilles tendon that may develop symptoms in a shorter duration of time, whereas tendinosis indicates a change in tendon structure that would require a longer duration of time for symptoms to develop (Scott et al., 2020). Additionally, the clinical condition of Achilles tendinopathy does not display the characteristics of an inflammatory response such as with tissue tearing (Cook et al., 2016), and the structural changes, as those expected in tendinosis, are not required to be present for pain or dysfunction to develop (Cook et al., 2016).

The nine core health domains of tendinopathy

Vicenzino et al. (2020) reported that the lack of agreed upon tendon health related domains impedes the progress of tendinopathy research. The nine identified domains (patient rating of overall condition, pain on activity or loading, participation, function, psychological factors, disability, physical function capacity, quality of life, and pain over a specified timeframe) should allow for greater consistency in the reporting of tendon research (Vicenzino et al., 2020). This scoping review further highlights the inconsistency in the methods used to diagnose and assess Achilles tendinopathy. There was variation in the methodology used to clinically diagnose and assess Achilles tendinopathy for all key themes; subjective history, clinical tests and outcome measures.

Subjective history

Multiple measures were identified to determine a diagnosis of Achilles tendinopathy from the subjective interview (Tables 4, 5, 6, 7, 8, 9, 10, 11 and 12). The most commonly used measure was self-reported location of pain, with midportion Achilles tendinopathy most commonly being defined as an area located 2–6 cm above the calcaneal insertion of the Achilles tendon. Insertional tendinopathy was most commonly defined as the distal 2 cm of the Achilles tendon. Additional measures included pain with tendon loading activity, duration of symptoms, and tendon stiffness following tendon loading or at a particular time of the day (i.e. morning stiffness). Interestingly, while a change in Achilles tendon loading activity (both an increase and decrease) is considered a catalyst for Achilles tendinopathy (Cook et al., 2016), it was only utilised as a specific criterion in nine of the included studies (Bains & Porter, 2006; Barker-Davies et al., 2017; Cook, Khan & Purdam, 2002; Courville, Coe & Hecht, 2009; Feilmeier, 2017; Maffulli et al., 2012; Nichols, 1989; Simpson & Howard, 2009; Sorosky et al., 2004).

Objective clinical tests

As with subjective history, numerous clinical tests were identified to diagnose Achilles tendinopathy (Tables 4, 5, 6, 7, 8, 9, 10, 11 and 12). The most commonly identified clinical test for Achilles tendinopathy was tendon palpation (including pain on palpation, localised tendon thickening or localised swelling). Although, palpation is commonly used to identify the region of pain and is a common clinical measure used to diagnose Achilles tendinopathy (Hutchison et al., 2013; Reiman et al., 2014; Maffulli et al., 2003; Martin et al., 2018), studies reported multiple regions of interest for midportion Achilles tendinopathy. Painful regions were described as the ‘midportion’, ‘middle third’, 2 to 4, 2 to 5, 2 to 6, 2 to 7, 4 to 6 and 4 to 7 cm above the calcaneal insertion. Similarly, the region of interest in insertional Achilles tendinopathy was described as the ‘insertion’, calcaneal tuberosity, distal 2 cm, and distal 5 cm.

While there was consistency in the included studies in their use of palpation as a clinical test, there is significant variation in the additional clinical tests used to confirm a diagnosis of Achilles tendinopathy (Tables 4, 5, 6, 7, 8, 9, 10, 11 and 12). Further clinical tests used to assess Achilles tendinopathy included tendon pain during loading activities (single-leg heel raises and hopping). The most frequently used clinical, tendinopathy specific tests, were the Royal London Hospital Test and the Painful Arc Sign. The Royal London Hospital Test is considered positive when the is a reduction in palpable Achilles tendon pain on ankle dorsiflexion (Maffulli et al., 2003). The Painful Arc Sign is considered positive when the area of swelling identified with palpation moves with active ankle plantarflexion and dorsiflexion (Maffulli et al., 2003).

Outcome measures

As was the case for the clinical features, there were significant variations in the outcome measures utilised for making a diagnosis of Achilles tendinopathy (Fig. 4). While overall disability and participant perceived pain were commonly measured, the impact of Achilles tendinopathy on quality of life and psychological factors were rarely measured. Psychological factors such as pain efficacy, catastrophisation and kinesiophobia scales were identified as important outcome measures in the diagnosis tendinopathy (Vicenzino et al., 2020), which aligns with the identified psychological outcome measures (PES, TKS and PCS). Similarly, disability measures that combine patient rated pain and function in relation to tendon-specific activities were identified as integral to monitoring tendinopathy outcomes. This scoping review identified the VISA-A questionnaire as the most commonly used outcome measure to monitor Achilles tendinopathy. In addition to psychological factors and disability, overall quality of life was identified as a core health domain in tendinopathy (Vicenzino et al., 2020). The scoping review identified three different outcome measures (SF-12, SF-36 and EQ-5D-5L) that were utilised to assess participant quality of life.

An example evidence-based method for clinically diagnosing Achilles tendinopathy

While there was significant variation in the methods used to diagnose Achilles tendinopathy, some common themes can be identified. When considering a consistent method for diagnosing and assessing Achilles tendinopathy, it is important to ensure research follows consensus recommendations on both terminology used and reporting outcomes (Scott et al., 2020; Vicenzino et al., 2020). Thus, Table 13 provides an amalgamation of the common features used to diagnose Achilles tendinopathy identified in the scoping review and the previously identified nine core health domains for tendinopathy (Vicenzino et al., 2020). While the VISA-A is Achilles tendon specific, any validated and reliable pain questionnaire and quality of life questionnaire may be used in place of the Pain Catastrophising Scale and SF-12.

Table 13 A method for clinically diagnosing Achilles tendinopathy.

Test	Definition of test	Feature	Core health domain	
Subjective history				
Self-reported location of pain	Clinician asks patient “Can you point out where you get your pain”	Pain located 2–6 cm above the calcaneal insertion (midportion)
Pain located in the distal 2 cm of the Achilles tendon	N/A	
Self-reported pain with tendon loading	Patient reported intensity of pain using a VAS or NPRS while performing an Achilles tendon-specific loading task (single-leg heel raise, hopping)	Patient reported increased pain on a VAS or NPRS with Achilles tendon-specific loading task (single-leg heel raise and hopping)	Pain with loading or activity	
Self-reported tendon stiffness or pain over a specified time	Clinician asks about pain and stiffness over specified timeframes (e.g. morning, night, 24 hours)	Patient reported morning stiffness or pain
Patient reported pain or stiffness at the onset of activity that may “warm-up”	Pain over a specified time	
Self-reported overall rating of Achilles tendon	Clinician asks “Can you rate your Achilles tendon where 100% represents no problems and 0% is the worst-case scenario”	Patient reported level of condition	Patient rating of overall condition	
Objective tests				
Palpation	Performed by the clinician gently palpating the whole length of the tendon in a proximal to distal direction	Patient reported pain located 2–6 cm above the calcaneal insertion (midportion) with or without subjective opinion of tendon thickening or swelling
Patient reported pain located in the distal 2 cm of the Achilles tendon with or without subjective opinion of tendon thickening or swelling	N/A	
Single-leg heel raise	Performed by patient rising up on to tip toes and lowering back down in a controlled manner, on both the affected and non-affected leg	Clinician recorded number of completed single-leg heel raises on each leg	Physical function capacity	
Hopping	Performed by participant hopping on the spot	Clinician recorded number of completed hops on each leg	Physical function capacity	
The Royal London Hospital Test	Performed by the clinician palpating the tendon for any local tenderness with the ankle either in neutral position or in slight plantarflexion. The ankle is then actively dorsiflexed and plantarflexed. With the ankle in maximum dorsiflexion, the portion of the tendon found to be tender is palpated again.	Patient reported pain on palpation reduces significantly or disappears with maximum dorsiflexion	N/A	
Painful Arc Sign	Performed by the clinician identifying the intratendinous swelling in the tendon and asking the patient to actively dorsiflex and plantarflex the ankle joint observing the movement of the swelling between the malleoli	The intratendinous swelling moves relative to the malleoli with the Achilles tendon during the ankle movement	N/A	
Outcome measures				
VISA-A	The VISA-A questionnaire is a valid and reliable tool to evaluate clinical severity of Achilles tendinopathy that has been translated into multiple languages. Patients can self-administer the questionnaire	The maximum score is 100, with healthy subjects scoring a minimum of 96 (Malliaras et al., 2015).	Disability	
PCS	Patients are asked to indicate the degree to which they have the above thoughts and feelings when they are experiencing pain using the 0 (not at all) to 4 (all the time) scale.	A total score is yielded (ranging from 0-52), with a score of 30 or below indicating a clinically relevant level of catastrophising (Scott et al., 2013).	Psychological factors	
SF-12	The SF-12 is a self-reported outcome measure assessing the impact of health on an individual’s everyday life.	The SF-12 creates two summary scores, mental health and physical health (Pham et al., 2014).	Quality of Life	
Note:

N/A, not applicable; cm, centimetres; VAS, Visual Analogue Scale; NPRS, Numerical Pain Rating Scale; VISA-A, Victorian Institute of Sport Assessment-Achilles; PCS, Pain Catastrophising Scale; SF-12, 12-Item Short Form Survey.

Limitations

This review was limited to publications in English, which may have excluded key studies published in other languages. Additionally, the screening, inclusion, exclusion and data extraction was performed by one reviewer (WM), which decreases the probability all relevant studies were identified for review and could lead to reviewer bias. The methodological quality of the studies was not assessed as per guidelines for completing scoping reviews (Peters et al., 2020; Arksey & O’Malley, 2005), meaning studies of poor design are given equal weighting to those of better quality, however, the descriptive nature of the scoping review limits the potential impact of individual studies’ methodological quality on results. The aim of a scoping review is to provide an overview of all literature within a field of evidence (Pham et al., 2014), and while there is no specific requirement for methodological quality appraisal, assessing individual literature methodological quality utilising a standardised tool may help authors identify gaps in the literature related to low quality research in addition to lack of research.

Conclusions

The specific objectives, including the most common clinical tests used to diagnose Achilles tendinopathy and identifying the most common outcome measures used to assess Achilles tendinopathy were highlighted, with the scoping review identifying the significant variation in the methodology and outcome measures used to diagnose Achilles tendinopathy. This scoping review provides a detailed summary of the current evidence and common themes were identified in the available research to provide an evidence-based method to diagnose Achilles tendinopathy utilising both subjective and objective testing, in addition to recommendations regarding common outcome measures. The primary aim of this scoping review was to identify and provide a method for clinically diagnosing Achilles tendinopathy that aligns with the nine core health domains and a method for diagnosing Achilles tendinopathy is proposed, that includes both results from the scoping review and recent recommendations for reporting results in tendinopathy. The development of a method for the clinical diagnosis of Achilles tendinopathy is key to developing greater homogeneity in future research. By standardising the clinical diagnosis of Achilles tendinopathy, future research is able to investigate other areas of this complex condition and identifying possible subclassifications of Achilles tendinopathy and thus improving tailored individual treatment programmes or monitoring patient progress. Additionally, an evidence-based method for the clinical diagnosis of Achilles tendinopathy will allow clinicians to be more confident with their diagnosis and provide patients with greater certainty.

Supplemental Information

Supplemental Information 1 Raw Data.

All collected data from all included studies used in the scoping review.

Click here for additional data file.

Supplemental Information 2 PRISMA Checklist.

Click here for additional data file.

Additional Information and Declarations

Competing Interests

Author Contributions

Data Availability

The authors declare that they have no competing interests.

Wesley Matthews conceived and designed the experiments, performed the experiments, analyzed the data, prepared figures and/or tables, authored or reviewed drafts of the paper, and approved the final draft.

Richard Ellis conceived and designed the experiments, authored or reviewed drafts of the paper, and approved the final draft.

James Furness conceived and designed the experiments, authored or reviewed drafts of the paper, and approved the final draft.

Wayne A. Hing conceived and designed the experiments, authored or reviewed drafts of the paper, and approved the final draft.

The following information was supplied regarding data availability:

The raw data is available in the Supplementary File.

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
