# Peer review of "The clinical diagnosis of Achilles tendinopathy: a scoping review"

_PeerJ, doi:10.7717/peerj.12166_

## Round 0.1 · original submission · Major Revisions

Authors, please, kindly see the feedback provided from the reviewers, who have found a lot of merits in your work. A reviewer considered acceptance, the other minor revisions. In addition, I encourage you to please consider the following important points to strengthen this important piece of work:

1) Introduction:
-Please, the second paragraph, separate from that starting line 61, should be dedicated to 'Explain the concept of diagnosis, and how does it differ from clinical diagnosis? What key things does one look out for in clinical diagnosis? Why is clinical diagnosis important in health situation of given patient. This paragraph should end with the sentence like : that is why it becomes relevant to understand the clinical diagnosis associated with, for instance Achilles tendinopathy.
This will then make paragraph starting line 61 very fitting, to a reader who is a learner in your specialty.
-The justification of this work is not very strong. Please, add more reflective gap analysis to build a strong justification, as to why this study is relevant? Why? Where? Which? How? Please brainstorm and provide more information in last but one paragraph, lines 72-79
- Line 80, please, it is not so right to say 'primary aims'. A study should have an aim. In that aim, there can be one, two, three or more 'specific objectives'. So, please, identify what is the overall aim? then, what are the specific objectives.

2) Methods:
Study design is not complete. Study design ought to show how the entire study layout is. Do you get it? Please, provide a schematic flow of how this entire study design lay out is. This will be your figure 1. This study design should begin with Figure 1 shows the schematic flow of the study design followed to actualise this study, beginning from ....., to .....
Succinctly describe this figure 1, in this study design section. This figure should show pathway of entire methodology, and must be in the context of addressing the specific objectives of this work.

3) The results section is very good. I have no comments there

4) Discussion:
Make the discussion its main heading, and others under it, subheadings, so that it clearly shows there are under the discussion.
-In the discussion, please authors, make every effort to indicate (refer to Table ??) or (Refer to Figure??) in all the places where specific Table or Figure mentioned in result section is being referred to in discussion. That is to say, all the Table(s)/Figure(s) mentioned in the results section, must be captured in the discussion.
The editor will be looking out for this in your revised manuscript.

5) Limitations is ok.However, it is important to brainstorm on what could be a way to overcome the challenge of methodological quality of the studies not being assessed as per guidelines for completing scoping reviews? Please brainstorm and include it from Line 458.

6) Conclusions: It will be important to reiterate how the review achieved its objectives. It will be important to add more information as to what makes this work relevant within the existing body of knowledge, and how it fills the gap. What are lessons for the general public? Please, also brainstorm the direction of future studies, which could be analytical, or even review.

This is a very scholarly work. The editor believes that by addressing all the above, in addition to those the reviewers have raised, the authenticity and quality of this work will dramatically increase.

I will diligently examine your revised manuscript, and therefore encourages authors to carefully address every single point raised herein.

Thank you very much.

Reviewer 1 ·

Basic reporting

Dear editor,
dear authors,
thank you for the invitation to review your paper, which I enjoyed a lot. The article structure, the literature reference and the study design had a high quality and your article will give good recommendations for researches for study designs and outcome parameters for future clinical studies in the field of Achilles tendinopathy

Experimental design

You have chosen a scoping review design which is totally appropriate.

Validity of the findings

Your conclusion are well stated and linked to the original research questions.

Additional comments

Here are some specific comments:

l.91. Full stop after „April 2021“.
l.119. „asymptomatic Achilles tendinopathy“ might be a misleading terminology. since „tendinopathy“ includes a dysfunctional or symptomatic state (see your definition ll. 55-58). Perhaps, „ asymptomatic Achilles tendon states“ might be more suitable.
l.164. „will“ instead of „with“

table1: As far as I understand you are referring to the study of Vicenzino et al. (8) so you would need to cite this at this point. However, since your study is about Achilles tendinopathy I would rather recommend to adapt this table and exclude unrelevant scores such as the DASH score.
Moreover, I would recommend to give researches at this point direct examples of scores or questionaires for every core health domain e.g. participation (Tegener activity scale) as you also named some in the discussion. Thus researaches will directly know what to do to improve their study design, when they have a look at table 1.

table 6: Why you did not include VISA-A here as you did e.g. in table 7( e.g. the study of Gatz et al. used VISA-A)?

Reviewer 2 ·

Basic reporting

The present manuscript was well structured with clearly results and conclusions of this current topic.

Just a few suggestions for the autors to improve tje quality of the manuscritp:

Please, add some studies about tendinopathy and ultrasound imaging (for example Romero-Morales et al. “Comparison of the sonographic features of the Achilles Tendon complex in patients with and without achilles tendinopathy: A case-control study”. Physical Therapy in Sport.) and more studies related with ultrasound imaging findings and tendinopathy.

Please, authors should write the conclusion section more “conclusive”, just with the main findings.

At last, congratulations for this excellent work.

Experimental design

Well designed.

Validity of the findings

No comment

---

## Round 0.2 · accepted · Accept

Authors, after a careful evaluation of your revised manuscript, the editor is very satisfied that all concerns raised have been addressed. The revised manuscript is now acceptable for publication. Thank you for finding PeerJ as your journal of choice. Look forward to your future scholarly contributions.

Congratulations and very best wishes